https://doi.org/10.1038/s41467-021-24302-6　　**OPEN**

# The long non-coding RNA βFaar regulates islet β-cell function and survival during obesity in mice

Fangfang Zhang[1,5], Yue Yang[1,5], Xi Chen[1], Yue Liu[1], Qianxing Hu[1], Bin Huang[1], Yuhong Liu[1], Yi Pan[1], Yanfeng Zhang[1], Dechen Liu[2], Rui Liang[3], Guoqing Li[2,4], Qiong Wei[2,4✉], Ling Li[2,4✉] & Liang Jin [1✉]

Despite obesity being a predisposing factor for pancreatic β-cell dysfunction and loss, the mechanisms underlying its negative effect on insulin-secreting cells remain poorly understood. In this study, we identify an islet-enriched long non-coding RNA (lncRNA), which we name β-cell function and apoptosis regulator (*βFaar*). *βFaar* is dramatically downregulated in the islets of the obese mice, and a low level of *βFaar* is necessary for the development of obesity-associated β-cell dysfunction and apoptosis. Mechanistically, *βFaar* promote the synthesis and secretion of insulin by upregulating islet-specific genes *Ins2*, *NeuroD1*, and *Creb1* through sponging *miR-138-5p*. In addition, using quantitative mass spectrometry, we identify TRAF3IP2 and SMURF1 as interacting proteins that are specifically associated with *βFaar*. We demonstrate that SMURF1 ubiquitin ligase activity is essential for TRAF3IP2 ubiquitination and activation of NF-κB-mediate β-cell apoptosis. Our experiments provide direct evidence that dysregulated *βFaar* contributes to the development of obesity-induced β-cell injury and apoptosis.

[1] State Key Laboratory of Natural Medicines, Jiangsu Key Laboratory of Druggability of Biopharmaceuticals, School of Life Science and Technology, China Pharmaceutical University, Nanjing, Jiangsu Province, China. [2] Department of Endocrinology, Zhongda Hospital, School of Medicine, Southeast University, Nanjing, Jiangsu Province, China. [3] Organ Transplant Center, Tianjin First Central Hospital, Nankai University, Tianjin, China. [4] Pancreatic Research Institute, Southeast University, Nanjing, China. [5] These authors contributed equally: Fangfang Zhang, Yue Yang. ✉email: weiqiong_seu@163.com; li-ling76@hotmail.com; ljstemcell@cpu.edu.cn

The rising incidence of type 2 diabetes (T2D) correlates with spiraling levels of obesity[1]. A common feature of T2D is elevated blood glucose and/or high serum free fatty acids (FFAs)[2]. Normal blood glucose homeostasis is dependent on the activity of insulin, which is secreted by the pancreatic β-cell in the islets of Langerhans[3]. The development of insulin resistance in response to obesity, causes an increase in plasma glucose levels and increases the demand on β-cell to produce and secrete insulin[4]. Prolonged exposure to excess glucose and FFAs negatively affects specialized functions of β-cells, decreasing the capacity to synthesize and release insulin in response to secretagogues, and ultimately to the loss of β-cell by apoptosis[5,6]. However, the underlying molecular mechanism remains elusive.

Emerging evidence demonstrates the essential function of long non-coding RNAs (lncRNAs) in the regulation of β-cell function and the potential contribution of this type of RNA to the pathogenesis of diabetes[7,8]. Based on transcriptome profiling studies of islets and β-cell, more than 1000 islet-specific lncRNAs have been identified in both human and mouse islets although only a few have been well characterized[9,10]. The possible role of lncRNA in the physiology and disorders of pancreatic islets was first identified by Ding and coworkers, who reported that lncRNA *H19* is involved in the pathogenesis of islet dysfunction after gestational diabetes[11]. In another investigation, Fadista and collaborators documented that lncRNA *LOC283177* expression was directly related to insulin exocytosis by co-expression with genes critical for normal islet function, such as *Pax6*, *Syt11*, and *Madd*[12]. Arnes and colleagues identified lncRNA *βlinc1* as a *cis* regulator of the islet transcription factor NKX2.2 and showed that *βlinc1* KO mice exhibit impaired glucose tolerance due to defects in insulin secretion[13]. The group of Akerman reported that lncRNA *PLUTO* affects local 3D chromatin structure and transcription of PDX1 and that both *PLUTO* and PDX1 are downregulated in islets from donors with T2D or impaired glucose tolerance[10]. Interestingly, some LncRNAs are responsive to obesity. Motterle and coworkers reported that the expression of *βlinc1* and *βlinc2* correlated with body weight gain and glucose levels in diabetic animals, and the expression of the human ortholog of *βlinc3* in the islets of T2D patients was altered and was related to the body mass index[14]. Also, our most recent work demonstrated that lncRNA *Roit* expression was downregulated in the islets of obese mice, impairing the transcription of the insulin gene and glucose homeostasis[15]. Although these studies provided an important groundwork documenting the function of lncRNAs in β-cell, how these lncRNAs contribute to obesity-mediated β-cell dysfunction and apoptosis remains to be determined.

Here, we identify a lncRNA enriched in islets, named the β-cell function and apoptosis regulator *(βFaar)*, which is necessary and sufficient for β-cell apoptosis and islet-specific gene induction. We demonstrate that *βFaar* is downregulated in the islets of db/db mice. Both in vitro and in vivo, downregulation of *βFaar* enhanced β-cell dysfunction and apoptosis. Furthermore, luciferase reporter and RNA pull-down assays used to explore the regulatory mechanism, demonstrated that *βFaar* functions as a competing endogenous RNA (ceRNA) by sponging *miR-138-5p* to upregulate the expression of target genes *Ins2*, *NeuroD1*, and *Creb1*. In addition, we pulled down *βFaar* and identified its direct interacting proteins using quantitative mass spectrometry. *βFaar* associates with E3 ubiquitin ligase SMURF1, as well as with its ubiquitination substrate TRAF3IP2. In this manner, *βFaar* facilitates the ubiquitination of TRAF3IP2 by SMURF1 in MIN6 cells and accelerates TRAF3IP2 degradation. *βFaar* levels are downregulated in the islets of db/db mice, causing rapid decay of TRAF3IP2, and promoting NF-κB mediated β-cell apoptosis. Collectively, *βFaar* is obesity responsive and appears as an essential regulator of β-cell biology. The downregulation of *βFaar*

links obesity enhancement to β-cell dysfunction and apoptosis. This study identifies a mechanism of obesity mediated β-cell failure induced by *βFaar* inhibition.

## Results

### *βFaar* has lower-than-normal expression in islets of obese mouse models.
To identify lncRNAs potentially contributing to the development of obesity-associated dysfunction and apoptosis of β-cell, we performed global lncRNA expression profiling in pancreatic islets obtained from db/db mice. These mice develop severe obesity-associated diabetes due to impaired β-cell function and augmented apoptosis. The characteristics of the animals used in this study are listed in Table S1. Hierarchical clustering analyses identified distinguishable lncRNA expression patterns. Using an absolute fold change of at least 1.5 and a *p* value of less than 0.05, we observed upregulation of 138 lncRNAs and downregulation of 306 lncRNAs in db/db mice compared with control mice (Fig. 1a left, Fig. S1a). The performed screen confirmed a decreased expression of lncRNAs previously associated with obesity-induced β-cell dysfunction such as *Pluto*[10], *Roit*[15], and *Ptpdr*[16]. We found that 14 lncRNAs were downregulated in islets of db/db mice more than 8-fold (Fig. 1a, right), these lncRNAs were annotated using UCSC and Ensemble.

In the present study, we focused on the lncRNA *E130307A14Rik* located on chromosome 10qB1 in mice and chromosome 6q21 in humans (Fig. S1b). Based on its function unveiled below, we named this lncRNA as β-cell function and apoptosis regulator *(βFaar)*. *Mus-βFaar* has only one variant and *Has-βFAAR* has four variants. The UCSC genome browser view of the *Mus-βFaar* shows location conserved with human *βFAAR* by PhyloP statistical model (Fig. S1b). And we found that *has-βFAAR* transcript variant 4 in the human islets showed a higher expression level (Fig. 1b and Fig. S1c). Translational analysis failed to find any significant open reading frames with translational propensity (Fig. S1d). *βFaar* expression is highly enriched in pancreatic islets of normal mice (Fig. 1c) and decreased progressively body weight gain (Fig. 1d). While the expression of *βFaar* was only significantly decreased in islets, white adipose tissue (WAT), muscle, and kidney in db/db mice compared to age-matched wid-type (WT) mice of the same genetic background (Fig. S1e). Decreased *βFaar* expression in obese mouse models was further confirmed by quantitative real-time PCR with reverse transcription (qRT-PCR), which documented a 3.3-fold decrease in *βFaar* expression in islets of HFD-fed mice and a 6.5-fold decrease in *βFaar* expression in islets of db/db mice (Fig. 1e, f). Moreover, we compared *βFaar* expression in primary islet to exocrine glands isolated from normal mice, revealing that *βFaar* expression was 4-fold higher in islets than in exocrine glands, indicating that islets represent the main source of *βFaar* expression in the pancreas (Fig. 1g). Immunofluorescence showed that *βFaar* expression was enriched in β cells (Fig. S1f), and subcellular fractionation demonstrated detected that *βFaar* was moderately more abundant in the cytoplasm than in the nucleus of MIN6 cells and primary islet cells (Fig. 1h). This finding was further supported by fluorescent in situ hybridization (FISH) in MIN6 cells and mice islet (Fig. 1i). The subcellular localization of *βFaar* did not change in the islets of obese mice (Fig. S1g).

### Obesity-induced reduced expression of *βFaar* via DNMT3a and DNMT3b.
Obesity can cause an increase in serum concentration of glucose, free fatty acids, and inflammatory cytokines[17]. To determine the possible reason for the changes in *βFaar* expression in the islets of obese mice, we exposed MIN6 cells and normal mouse islets to glucose, pathophysiological concentrations of palmitate, and pro-inflammatory cytokines. The expression of *βFaar* decreased in the presence of palmitate

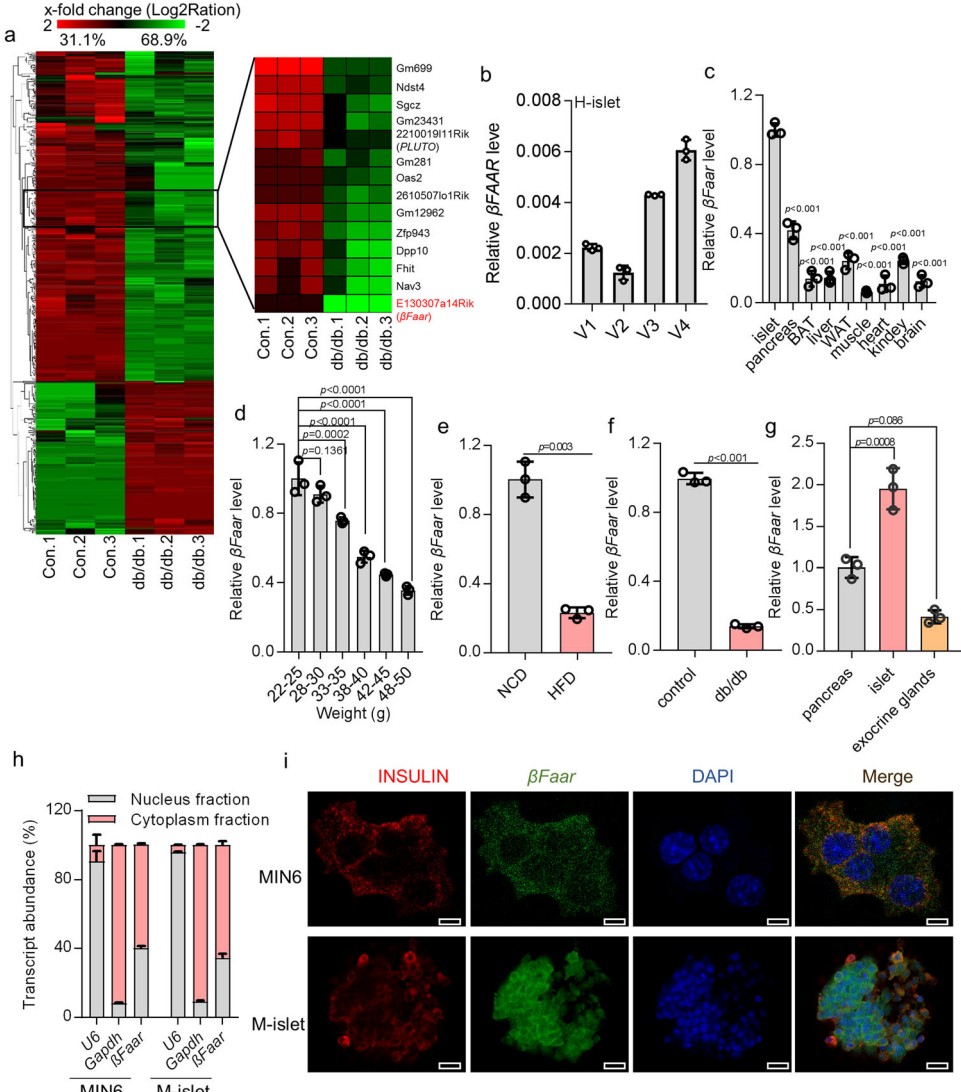

**Fig. 1 βFaar has lower-than-normal expression in islets of obese mouse models. a** Total islet RNA harvested from db/db mice ($n = 5$) and wide type mice ($n = 7$) was screened by RNA-sequence analysis. Microarray heatmap of differentially expressed lncRNAs. **b** qRT-PCR was performed to test the βFAAR (V1–V4) expression in the human islets. **c** The expression levels of *βFaar* in different tissues of C57BL/6J mice compared to islet ($n = 5$). White adipose tissue (WAT), brown adipose tissue (BAT). **d** The expression levels of *βFaar* in the islets at different stages (after 0-week, 4-week, 6-week, 8-week, and 16-week feeding HFD) during the development of obesity inducing diabetes ($n = 5$). The expression levels of *βFaar* in HFD mice (**e**, $n = 5$) and in db/db mice (**f**, $n = 5$). **g** The expression levels of *βFaar* in the pancreas, islets and exocrine glands ($n = 3$). Pancreas was digested by collagenase VI, all islet were picked out, other cells were exocrine glands. **h** *βFaar* was enriched in the MIN6 cell and primary islet cell cytoplasm fraction. **i** FISH analysis of *βFaar* in MIN6 cells and islet cells. The nuclei were stained with DAPI. Magnification: ×40 or ×20, scale bar, 10 μm or 20 μm. All experiments above were performed in triplicates, and each group contained three batches of individual samples. The p-values by two-tailed unpaired Student's t test (**e**, **f**), one-way ANOVA (**c**, **d**, **g**) are indicated. Data represent the mean ± SD. Source data are provided as a Source data file.

(0.5 mmol/l) (Fig. 2a) and pro-inflammatory cytokines (Fig. 2b), but not upon incubation with high glucose (33.3 mmol/l) (Fig. S2a). The same results were also observed in the human islets (Fig. 2c, d, and Fig. S2b).

LncRNA expression is often regulated by DNA methylation[18], and obesity can trigger epigenetic modifications[19]. This possibility prompted us to investigate whether the expression of *βFaar* is regulated by methylation. Using the MethPrimer database, we identified one CpG island located at the putative transcription start sites of *βFaar*, suggesting that DNA methylation might control the transcriptional activity of *βFaar* (Fig. S2c). Next, we found that the treatment of MIN6 cells and mice islets with 5-Aza-dC increased the expression of *βFaar* in a dose-dependent manner (Fig. 2e). To further investigate whether silencing of

*βFaar* in obese mice was caused by DNA hypermethylation, we assessed the extent of CpG methylation within the *βFaar* promoter in the islets of db/db mice by bisulfite sequencing. We found that the hypermethylation of CpG island in *βFaar* promoter in is significantly higher in the db/db *mice* than in control animals (Fig. 2f). Likewise, MIN6 cells treated with 0.5 mmol/l palmitate had a higher degree of methylation within the *βFaar* promoter than control cells (Fig. S2d).

DNA methylation in mammals is regulated by DNA methyltransferases (DNMTs), including DNMT1, DNMT3a, and DNMT3b[20–22]. To verify whether *βFaar* methylation is regulated by DNMTs, we examined the expression pattern of DNMTs in the islets of obese and normal mice. We found that the protein levels of DMNT3a and DMNT3b were significantly

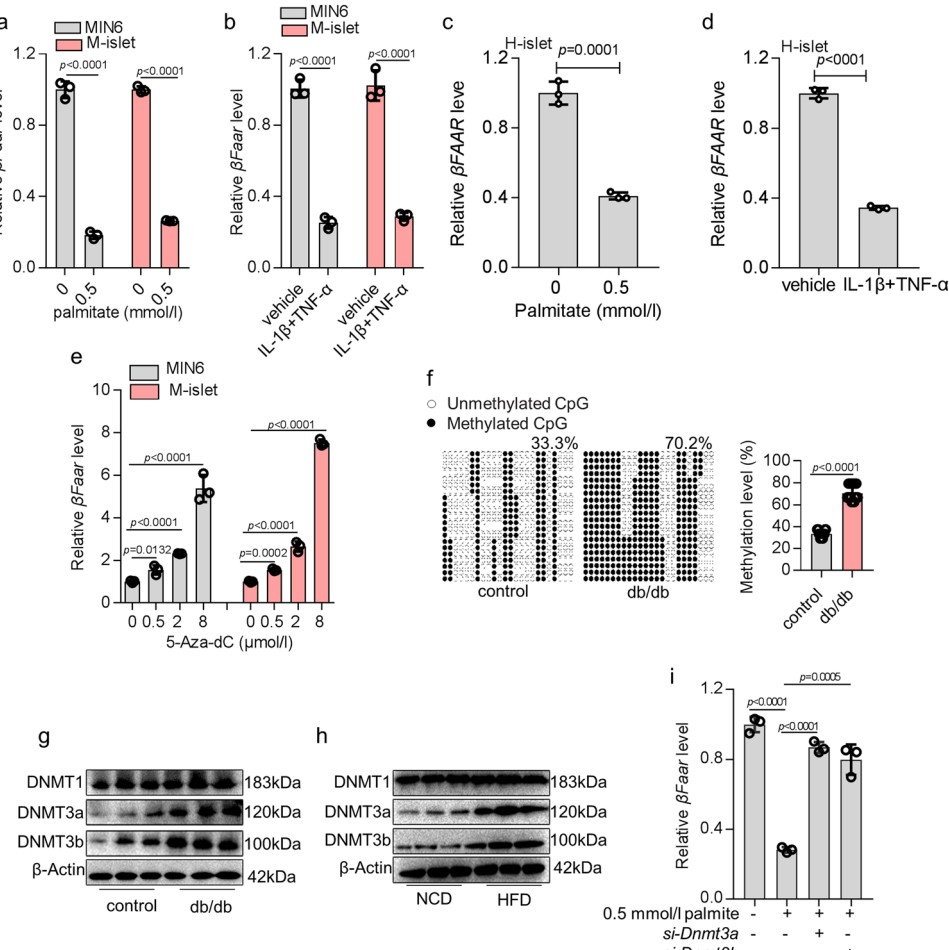

**Fig. 2 Obesity-induced reduced expression of *βFaar* via DNMT3a and DNMT3b.** MIN6 cells and mice primary islets were incubated with 0.5 mmol/l palmitate (**a**) and a combination of interleukin-1β (IL-1β, 5 ng/ml) and tumor necrosis factor-α (TNF-α, 30 ng/ml, **b**) for 48 h and qRT-PCR was performed to examine the *βFaar* levels. The expression levels of *βFAAR* in human islets incubated with 0.5 mmol/l palmitate (**c**) and a combination of interleukin-1β (IL-1β, 5 ng/ml) and tumor necrosis factor-α (TNF-α, 30 ng/ml, **d**), **e** MIN6 cells were incubated with different doses of 5-Aza-dC (0.5 μmol/l, 2 μmol/l, and 8 μmol/l) and qRT-PCR was performed to analyze the expression levels of *βFaar*. **f** One BSP regions of the *βFaar* promoter CpG island in the islet of db/db mice (n = 6). Each box indicates the methylation status of the CpG site. Each row represents an individual sequenced DNA strand. Over 20 clones from each mixed sample were sequenced. The percentage of methylation in each sequenced region was indicated. **g–h** The protein levels of DNMT1, DNMT3a, and DNMT3b in the islet of db/db mice (**g**) and HFD mice (**h**, n = 5). **i** MIN6 cells were incubated with 0.5 mmol/l palmitate and co-transfected with si-Dnmt3a or si-Dnmt3b, followed by qRT-PCR to examine the expression levels of *βFaar*. All experiments above were performed in triplicates, and each group contained three batches of individual samples. The p-values by two-tailed unpaired Student's t test (**c**, **d**, **f**), one-way ANOVA (**i**) or two-way ANOVA (**a, b**, **e**) are indicated. Data represent the mean ± SD. Source data are provided as a Source data file.

higher in the islets of diabetic db/db mice and HFD-fed mice than in controls, but the protein level of DNMT1 was not significantly changed (Fig. 2g, h). The rescue experiment demonstrated that palmitate-induced downregulation of *βFaar* was reversed when the mice islets were transfected with *si-Dnmt3a* and *si-Dnmt3b* (Fig. 2i). These results indicate that obesity-induced down-regulation of *βFaar* in the islets is mediated partly via *DNMT3a* and *DNMT3b*.

**Suppression of *βFaar* expression decreases insulin transcription and secretion.** To explore the potential role of *βFaar* in regulating β-cell function, we used smart silence targeting knockdown of *βFaar* (*si-βFaar*). This protocol involved a mixture of three siRNAs and three antisense oligonucleotides to knockdown *βFaar* in the cytoplasm and nucleus, respectively, a scrambled sequence (si-NC) was employed as a control. In addition, an overexpression plasmid (*oe-βFaar*) was used to upregulate *βFaar*. Mice islets and MIN6 cells were transfected with *si-βFaar*, si-NC,

*oe-βFaar*, or pcDNA3.1 vector (vehicle) and collected 48 h later. The efficiency of knockdown and overexpression were approximately 80% and 200-fold, respectively (Fig. S3a, b). *oe-βFaar* significantly increased the mRNA levels of insulin genes (*Ins1* and *Ins2*) (Fig. 3a, Fig. S3c) and the insulin content (Fig. 3b, Fig. S3d), Next, we performed glucose challenge experiments using mice islets and MIN6 cells with knockdown or overexpression of *βFaar*. The downregulation of *βFaar* decreased insulin secretion after exposure to high glucose, and insulin secretion was markedly increased when *βFaar* was overexpressed (Fig. 3c, Fig. S3e), while knockdown produced opposite results. Changes in *βFaar* expression did not affect the content of glucagon (Fig. S3f). And *βFAAR* was overexpressed or knockdown in human islets (Fig. S3g), we also found that overexpression of *βFAAR* in human islets could increase INSULIN gene expression (Fig. 3d), insulin content (Fig. 3e) and insulin secretion (Fig. 3f). Moreover, we assessed glucose-stimulated insulin secretion (GSIS) in islets from diabetic db/db mice. As shown in Fig. 3g, insulin secretion in response to

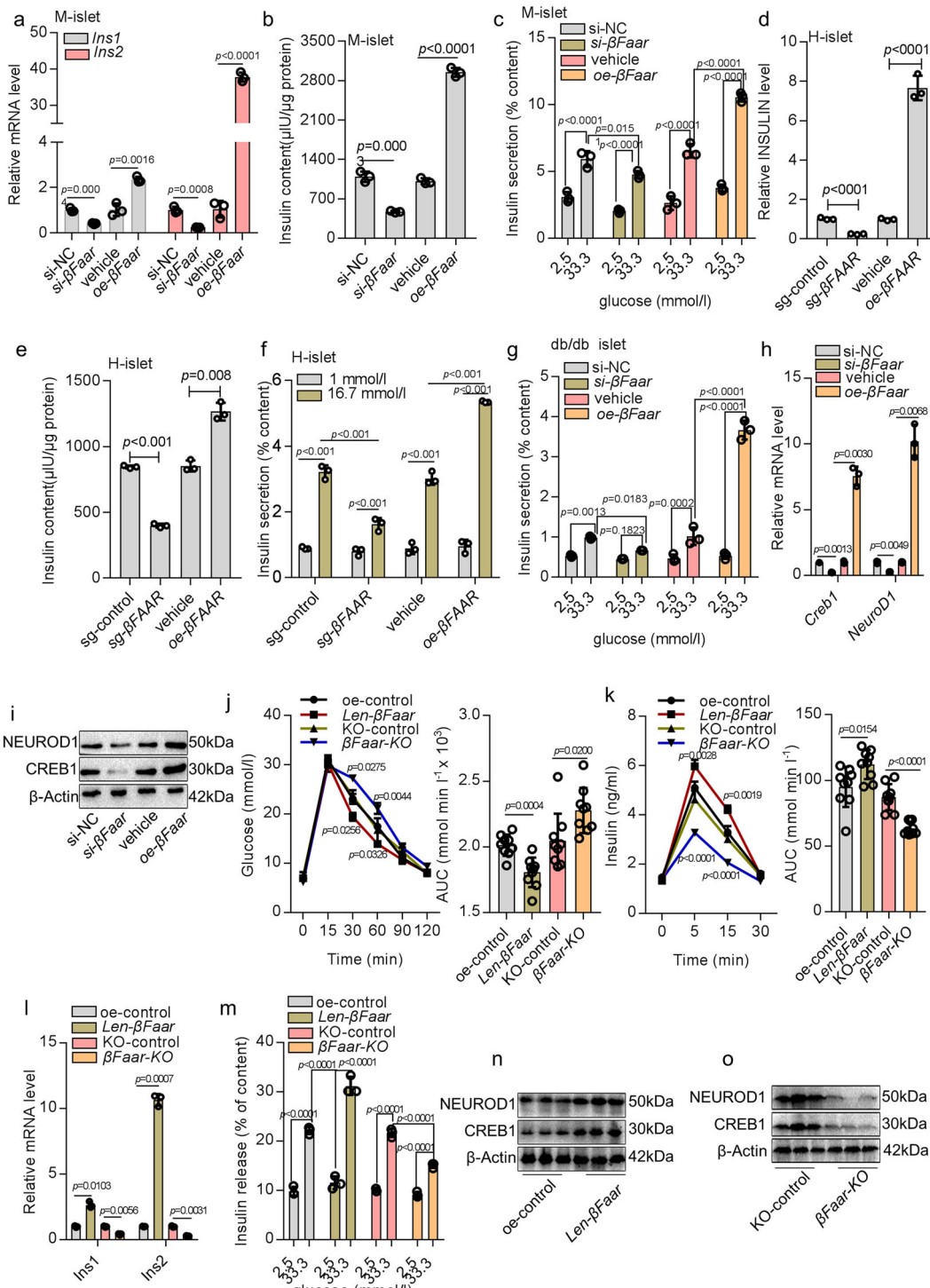

**Fig. 3 Suppression of *βFaar* expression decreases insulin transcription and secretion.** *βFaar* overexpression plasmid (*oe-βFaar*) and *βFaar* smart silence (*si-βFaar*) were transfected into mice islet cells for 48 h. Then mRNA levels of insulin genes (*Ins1* and *Ins2*, **a**, $n = 3$) were tested by qRT-PCR, insulin content (**b**, $n = 5$) and insulin secretion was analyzed by ELISA (**c**, $n = 5$). INSULIN gene expression (**d**), insulin content (**e**) and insulin secretion (**f**) in the human islets transfected with *oe-βFAAR* or *sg-βFAAR*. Insulin secretion in the primary islets of db/db mice transfected with *oe-βFaar* or *si-βFaar* (**g**, $n = 5$, 10 weeks old). The mRNA (**h**) and protein (**i**) level of CREB1 and NEUROD1. **j** IPGTT (2 g kg⁻¹) in overnight fasted *Len-βFaar mice* and control mice ($n = 9$). **k** In vivo insulin excursions in overnight fasted *Len-βFaar mice* and control mice after IPGTT exposure ($n = 9$). **l** Relative *Ins1* and *Ins2* expression levels in vivo ($n = 3$). **m** Static insulin secretion of islets ($n = 5$) at indicated glucose concentrations. **n–o** The protein levels of CREB1 and NEUROD1 in the islets of *Len-βFaar mice* (**n**) or *βFaar-KO* mice (**o**) ($n = 5$). All experiments above were performed in triplicates, and each group contained three batches of individual samples. The *p*-values by one-way ANOVA (**b**, **d**, **e**), or two-way ANOVA (**a**, **c**, **f–h**), and (**j–m**) are indicated. Data represent the mean ± SD, except (**j**, **k**) (mean ± SEM). Source data are provided as a Source data file.

high glucose was markedly enhanced when *βFaar* was over-expressed. To explain how downregulation of *βFaar* impairs insulin production, we measured both mRNA and protein expression of selected transcriptional factors with well-established function in insulin transcription including pancreatic and duodenal homeobox factor 1 (*Pdx1*), NK6 homeobox 1 (*Nkx6.1*), cAMP responsive element binding protein 1 (*Creb1*), paired box gene 6 (*Pax6*), neurogenic differentiation factor 1 (*NeuroD1*), and v-maf musculoaponeurotic fibrosarcoma oncogene homolog A (MafA). *βFaar* suppression selectively decreased CREB1 and NEUROD1 at both mRNA and protein levels (Fig. 3h, i). However, neither the mRNA nor the protein levels of PDX1, NKX6.1, and MAFA were affected by *βFaar* inhibition (Fig. S3h, i). These results suggest that the mechanism responsible for the decrease in insulin content may operate at the transcriptional level.

To verify whether ectopic expression of *βFaar* also affects insulin transcription and secretion in vivo, $1 \times 10^9$ lentivirus particles encoding *βFaar* (*len-βFaar*) or $1 \times 10^9$ lentivirus of *βFAAR* sgRNA, a dual CRISPR/Cas9 system to knock out the exogenous expression of *βFaar* (*βFaar-KO*) (Fig. S3j) was injected into 8-week old male C57BL/6 mice by the pancreatic intra-ductal infusion. We observed ~80-fold upregulation of *βFaar* in the islets that received *len-βFaar* compared to those receiving lentivirus pHAGE-CMV-Rat Insulin promoter (RIP) (oe-control). Moreover, there was approximately an 80% decrease in expression of *βFaar* in the islets of *βFaar-KO* mice compared to mice treated receiving lentivirus lentiCRISPRv2-Rat Insulin Promoter (RIP)-sgRNA (KO-control) (Fig. S3k). Body weight and blood glucose were similar in both treated and control groups of mice ($n = 9$) (Fig. S3l, m). In contrast, *len-βFaar* mice exhibited improved results of the intraperitoneal glucose tolerance test, while *βFaar-KO* showed impaired glucose tolerance ($n = 9$) (Fig. 3j). Importantly, insulin level at 5 and 15 min after glucose injection was higher in *len-βFaar* mice than in control animals ($n = 4$) (Fig. 3k). In addition, insulin synthesis and insulin sensitivity to glucose were increased in the islets derived from *len-βFaar* mice ($n = 5$) (Fig. 3l, m), while an opposite result was obtained in *βFaar-KO* mice. The ectopic expression of *βFaar* did not change the islet structure and the content of glucagon (Fig. S3n, o). Consistent with these findings, islets from the *len-βFaar* group showed an increased level of CREB1 and NEUROD1 mRNA (Fig. S3p) and protein, while these levels were decreased in *βFaar-KO* mice (Fig. 3n, o). These findings imply that downregulation of *βFaar* expression is responsible, at least in part, for the obesity-induced β-cell dysfunction.

**Inhibition of *βFaar* expression results in β-cell apoptosis**. To identify the effect of *βFaar* on maintaining β-cell mass, we modified its expression in MIN6 cells. The CCK-8 assay showed that silencing *βFaar* reduced cell viability (Fig. S4a). Moreover, the downregulation of *βFaar* in MIN6 cells, mimicking the conditions encountered in obesity, increased the number of apoptotic cells (Fig. 4a). Similar results were obtained by the TUNEL assay in primary cultures of mice islets, in human islets and in MIN6 cells (Fig. 4b, c and Fig. S4b). Furthermore, Western blotting documented changes in the expression of apoptosis-related proteins siRNA-treated cells, consistent with the flow cytometric and TUNEL analysis. The suppression of *βFaar* expression increased the level of cleaved caspase-3 and BAX and decreased the level of procaspase-3 and BCL-2; the over-expression of *βFaar* induced opposite effects (Fig. 4d). *βFaar* overexpression in MIN6 cells resulted in a striking reduction in palmitate-induced apoptosis, as assessed by annexin V staining and by counting the cells displaying pyknotic nuclei, further

suggesting that *βFaar* downregulation mediates the obesity-induced apoptosis of β-cell (Fig. S4c).

To verify whether downregulation of *βFaar* also causes β-cell apoptosis in vivo, we mimicked the obesity-associated decrease in *βFaar* expression by knocking down *βFaar* in normal C57BL/6J mice through the pancreatic ductal infusion of *βFaar-KO*. The downregulation of *βFaar* significantly increased the level of cleaved caspase-3 and BAX while decreasing the level of procaspase-3 and BCL-2 expression (Fig. 4e). Opposite effects were achieved with the overexpression of *βFaar* (Fig. 4f). Furthermore, the inhibition of *βFaar* in normal mice led to an increase in the number of TUNEL-positive β-cells (Fig. 4g). Next, we assessed the contribution of increased *βFaar* expression to the protection of islets by overexpressing *βFaar* in normal mice. β-cell mass in mice treated with *len-βFaar* was slightly higher than in control mice (Fig. S4d). These findings indicate that the knockdown of *βFaar* alone is sufficient to impair the function of β-cell and induce their apoptosis.

**Overexpression *βFaar* protects against HFD-induced β-cell dysfunction**. To test whether overexpression of *βFaar* can alleviate obesity-induced β-cell dysfunction, $1 \times 10^9$ lentivirus particles encoding *βFaar* (*len-βFaar*) were injected into 8-week old male C57BL/6 mice via the pancreatic ductal infusion, and then fed the mice with HFD for 16 weeks. The time pattern of *len-βFaar* injection and HFD feeding is shown in Fig. 5a. We observed about 80-fold upregulation of *βFaar* expression in the islets of mice treated with *len-βFaar* compared to those receiving empty vector, this difference persisted up to 16 weeks after the injection (Fig. 5b). The expression of *βFaar* was not significantly changed in other organs (Fig. S5a). *len-βFaar* treatment had no effect on cumulative energy intake (Fig. S5b), body weight (Fig. S5c), body fat content (Fig. S5d), adipocyte size (Fig. S5e). Overexpression of *βFaar* also did not affect on random-fed glycemia (Fig. S5f). However, the obesity-associated rise in serum insulin concentrations was reduced in HFD-fed animals treated with *len-βFaar* (Fig. 5c). Homeostatic model assessment of insulin resistance (HOMA-IR) values were significantly decreased in mice overexpressing *βFaar* (Fig. 5d). Consistent with this result, glucose tolerance tests revealed an improvement of glucose tolerance (Fig. 5e) and insulin sensitivity (Fig. 5f) upon *βFaar* overexpression. Moreover, we isolated islets of *len-βFaar* treated and control mice after 8 and 16 weeks of HFD feeding. GSIS results revealed that insulin release was markedly improved in mice treated with *len-βFaar* when islets were exposed to 33.3 mmol/l glucose (Fig. 5g, h). The fraction of TUNEL-positive β-cell in *βFaar*-overexpressing mice was lower than in control animals (Fig. 5i). Collectively, these results indicate that over-expression of *βFaar* in HFD-induced obese mice improves β-cell function and inhibits β-cell apoptosis.

**_βFaar_ regulates insulin transcription by sponging *miR-138-5p*.** lncRNAs may function as competing endogenous RNAs (ceRNAs) to sponge miRNAs, thereby derepressing the targets of miRNA and imposing an additional level of post-transcriptional regulation[23,24]. To identify the potential miRNA targets of *βFaar*, in silico analysis was performed using the LncBase$_{2.0}$ and RegRNA$_{2.0}$ databases, and identified four miRNAs that may act as biological targets of *βFaar* (Fig. 6a). According to the assumed function of ceRNA, the expression of lncRNA and its target miRNA should show a negative correlation[25]. Therefore, we tested the expression patterns of the four candidate miRNAs in the islets of db/db mice. The level of all four miRNAs increased in the islets of obese mice, as opposed to the downregulation of *βFaar* (Fig. 6b). The direct binding between the four miRNAs and *βFaar* was validated by the affinity pull-down

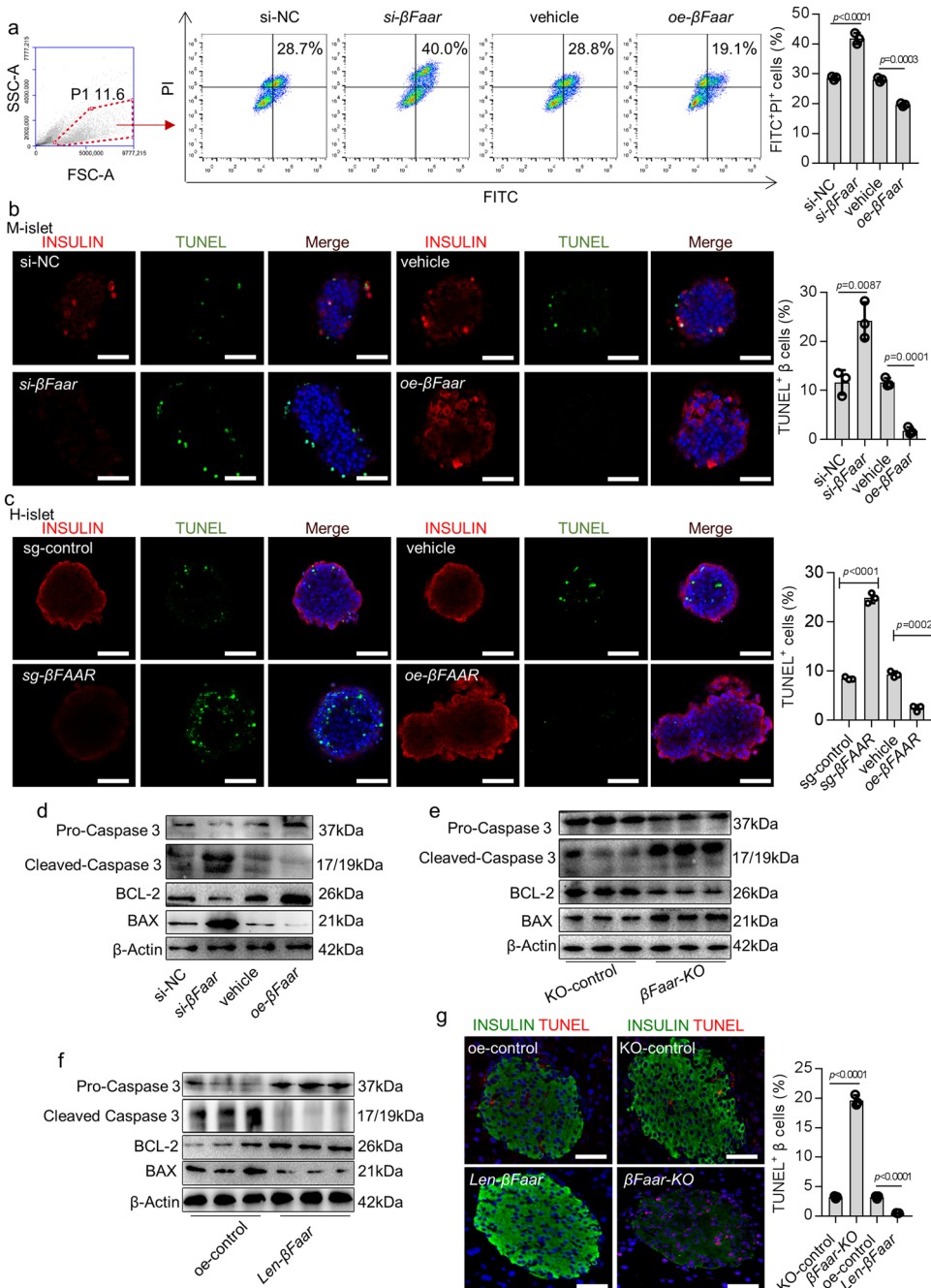

**Fig. 4 Inhibition of *βFaar* expression results in β-cell apoptosis.** *βFaar* overexpression plasmid (*oe-βFaar*) and *βFaar* smart silence (*si-βFaar*) were transfected into MIN6 cells for 48 h. Then the FITC⁺/PI⁺ cells were measured by Flow cytometry (**a**). **b**, **c** The both INSULIN and TUNEL positive cells were detected by immunofluorescence in the mice islets (**b**) and in the human islets (**c**). Magnification: ×40 or ×20, scale bar, 10 μm or 20 μm. **d** The protein levels of Pro-Caspase 3, Cleaved Caspase 3, BCL-2, and BAX in the MIN6 cells after transfection for 48 h. **e**, f The protein levels of Pro-Caspase 3, Cleaved Caspase 3, BCL-2, and BAX in the islet of *βFaar-KO* mice (**e**, n = 5) and *len-βFaar* mice (**f**, n = 5). **g** The TUNEL positive β cells in the islets of *βFaar-KO* mice and *len-βFaar* mice. Magnification: ×20, scale bar, 20 μm. All experiments above were performed in triplicates, and each group contained three batches of individual samples. The *p*-values by one-way ANOVA (**a**–**c**, **g**). Data represent the mean ± SD. Source data are provided as a Source data file.

of endogenous miRNAs associated with *βFaar* using in vitro transcribed biotin-labeled *βFaar* and demonstrated via qRT-PCR analysis. *βFaar-WT* in MIN6 cells was significantly enriched for *miR-138-5p* compared to blank (Beads), nontargeting micro-RNA (*miR-802-5p*), *βFaar* with mutations in miRNAs targeting sites (*βFaar-MUT*), and another *lncRNA Roit*, which is also downregulated in obesity but does not have a predicted miRNA targeting site (Fig. 6c). *βFaar-WT* was only slightly enriched for *miR-3099-5p* and *miR-1950* and did bind *miR-693-3p* (Fig. 6c).

*miR-138-5p* was selected for subsequent experiments based on the highest enrichment. The predicted binding sites of *miR-138-5p* to *βFaar* are illustrated in Fig. S6a. To further demonstrate the interaction between *βFaar* and *miR-138-5p*, we constructed luciferase reporters containing the 3' end 600 nt of *βFaar* with wild-type (WT) or mutated *miR-138-5p* binding sites. We found that the overexpression of *miR-138-5p* reduced the luciferase activity of the WT reporter vector but not the empty vector or mutant reporter vector (Fig. 6d). Moreover, an RNA pull-down assay using

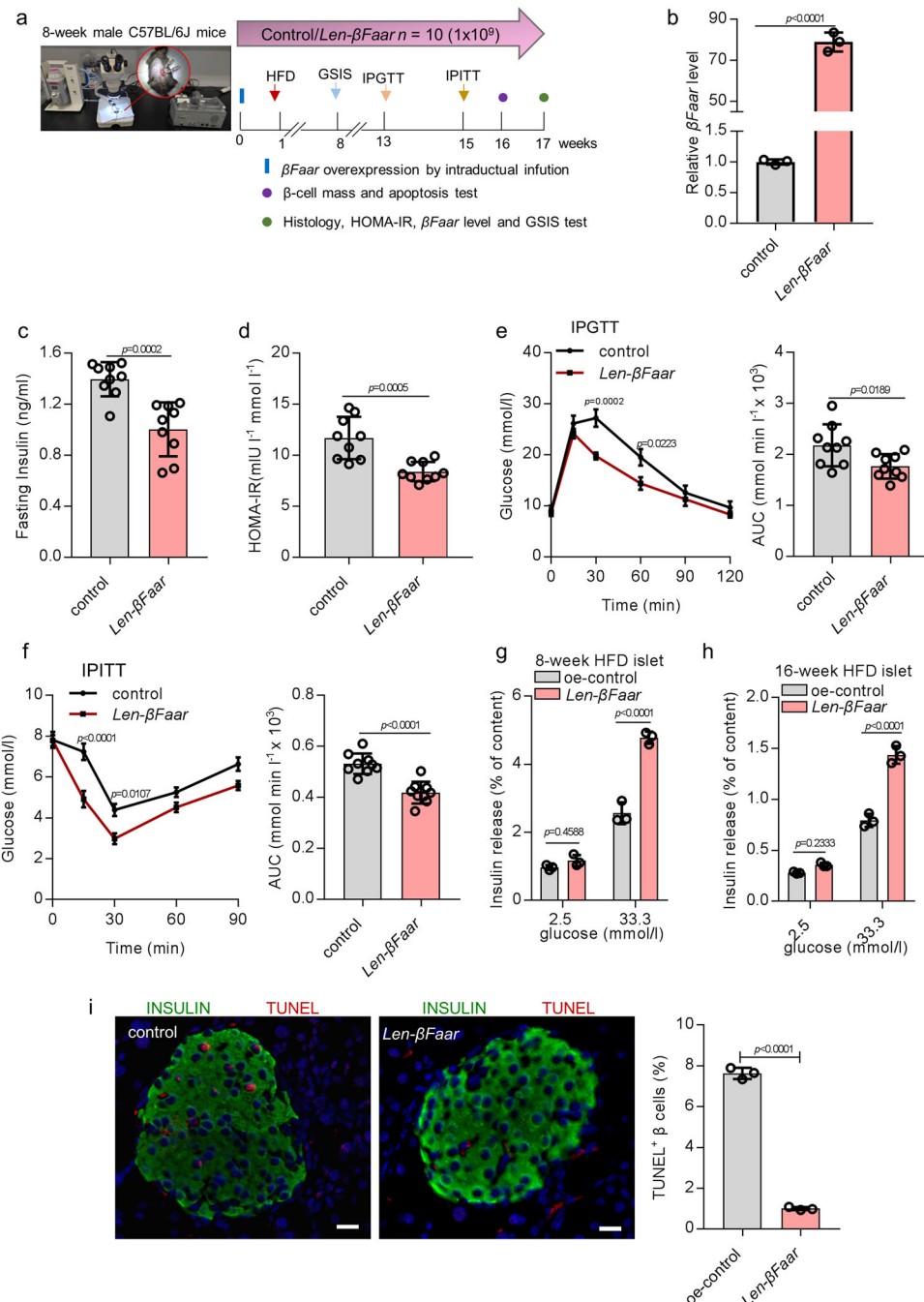

**Fig. 5 Overexpression βFaar protects against HFD-induced β-cell dysfunction. a** Flowchart of the in vivo experiments designed for detecting β-cell function via pancreatic ductal infusion ($n = 10$), 8-week-old male len-βFaar mice and control mice (Lentivirus pHAGE-CMV-Rat Insulin Promoter (RIP)) were exposed to HFD for 16 weeks. Then, the expression level of βFaar in the islets was measured by qRT-PCR (**b**, $n = 3$), fasting insulin levels (FINS) of HFD-fed mice were measured by ELISA (**c**, $n = 10$), the level of Homeostatic model assessment indices of insulin resistance (HOMA-IR) (**d**, $n = 10$). HOMA-IR was calculated with the equation (FBG (mmol $l^{-1}$) × FINS (mIU $l^{-1}$))/22.5. **e, f** Intraperitoneal glucose tolerance test (IPGTT) (2 g/kg) (**e**) and intraperitoneal insulin tolerance test (IPITT; 0.75 U/kg) (**f**) were performed in len-βFaar mice and control mice at the 12th or 14th week of High fat diet administered, respectively. The corresponding area under the curve (AUC) of blood glucose level was calculated ($n = 10$). **g, h** Insulin release from islets of len-βFaar mice or control mice after 8-week (**g**) or 16-week (**h**) HFD treatment ($n = 5$). **i** The TUNEL positive β cells were detected by immunofluorescence after 16-weeks' HFD treatment. Scale bar, 20 μm. All experiments above were performed in triplicates, and each group contained three batches of individual samples. The p-values by two-tailed unpaired Student's t test (**b–d**), and (**i**), or two-way ANOVA (**e–h**) are indicated. Data represent the mean ± SD, except (**c–f**) (mean ± SEM). Source data are provided as a Source data file.

biotin-labeled *miR-138-5p* revealed that *βFaar* was pulled down by *miR-138-WT*, whereas *miR-138-MUT* with a disrupted putative binding sequence failed to co-precipitate with *βFaar* (Fig. S6b). However, we found no significant difference in *βFaar* levels after

overexpression of *miR-138-5p* in MIN6 cells (Fig. S6c). Ectopically expressed *βFaar*, but not *lncRNA Roit*, reduced the levels of *miR-138-5p* (Fig. S6d). These data demonstrate that *miR-138-5p* binds to *βFaar* but does not trigger its degradation.

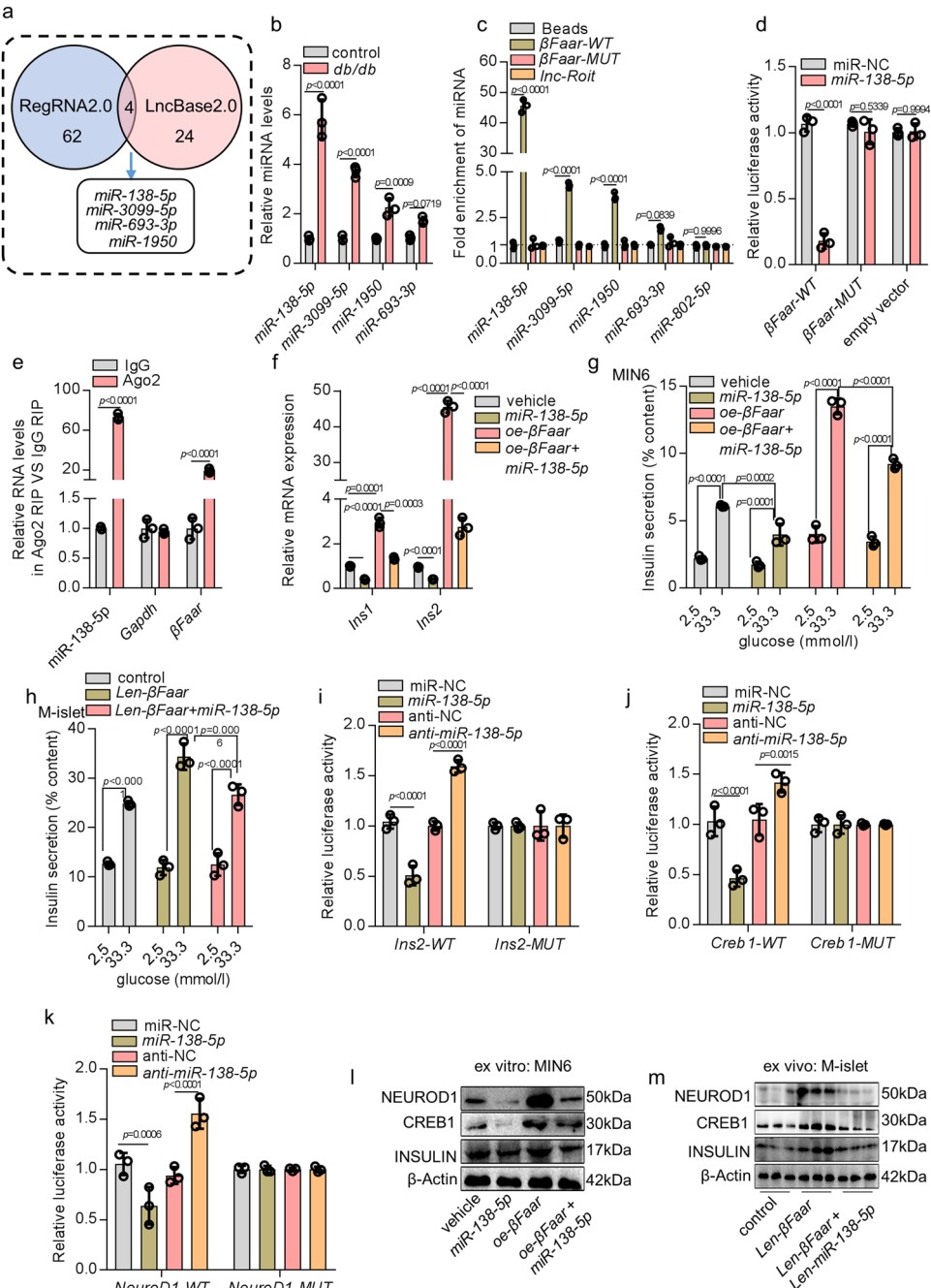

**Fig. 6 βFaar regulates insulin transcription by sponging miR-138-5p. a** RegRNA2.0 and LncBase2.0 were used to predict the target miRNA of βFaar, **b** qRT-PCR was performed to detect the expression levels of miR-138-5p, miR-3099-5p, miR-1950, and miR-693-5p in the islets of db/db mice. **c** MIN6 cell lysates were incubated with biotin-labeled βFaar, after pull-down, microRNAs were extracted and assessed by qRT-PCR. Nontargeting microRNA (miR-802-5p) and lncRNA Roit are also inhibited by obesity, but does not have a predicated miRNA targeting site. **d** Relative luciferase activity in MIN6 cells co-transfected with a luciferase reporter containing either βFaar-WT or βFaar-MUT (miR-138-5p-binding sequence mutated) and miR-138-5p mimics. Data are presented as the relative ratio of renilla luciferase activity and firefly luciferase activity. **e** Anti-Ago2 RIP was performed in MIN6 cells transiently overexpressing miR-138-5p, followed by qRT-PCR to detect βFaar associated with Ago2 (non-specific IgG as negative control). **f, g** miR-138-5p or βFaar was overexpressed in the MIN6 cells, qRT-PCR was performed to detect the Ins1 and Ins2 levels (**f**) and ELISA was carried out to test the insulin secretion (**g**). The insulin secretion in the islets of len-βFaar treated mice or both len-miR-138-5p and len-βFaar treated mice (**h, n** = 5). Relative luciferase activity in MIN6 cells co-transfected with a luciferase reporter containing either Ins2-WT or In2-MUT (**i**), Creb1-WT or Creb1-MUT (**j**), NeuroD1-WT or NeuroD1-MUT (**k**) and miR-138-5p mimics. Data are presented as the relative ratio of renilla luciferase activity and firefly luciferase activity. The protein levels of INSULIN, CREB1, and NEUROD1 in the MIN6 cells transfected with miR-138-5p mimics or βFaar overexpression plasmid (**l**) and in the islet cells of len-βFaar treated mice or both len-miR-138-5p and lent-βFaar treated mice (**m, n** = 5). All experiments above were performed in triplicates, and each group contained three batches of individual samples. The p-values by two-way ANOVA (**b**–**k**) are indicated. Data represent the mean ± SD. Source data are provided as a Source data file.

The miRNAs are known to bind their targets and cause translational repression and/or RNA degradation in an Ago2-dependent manner. To determine whether *βFaar* is regulated by *miR-138-5p* in such a manner, we conducted anti-Ago2 RIP in MIN6 cells transiently overexpressing *miR-138-5p*. Endogenous *βFaar* pull-down by Ago2 was specifically enriched in *miR-138-5p*-transfected cells (Fig. 6e), further supporting that *miR-138-5p* are bona fide *βFaar* targeting miRNAs. All these data demonstrate that *βFaar* physically associates with *miR-138-5p* and may function as a ceRNA.

To investigate whether *βFaar* regulates β-cell function through sponging *miR-138-5p*, a rescue experiment was performed by ectopic expression of *miR-138-5p*. While the overexpression of *βFaar* upregulated *Ins1* and *Ins2* transcripts, ectopic expression of *miR-138-5p* abrogated this increase (Fig. 6f). The promotion of insulin secretion in MIN6 cells or normal mice stably overexpressing *βFaar* after the administration of len-*βFaar* was abolished by the overexpression of *miR-138-5p* (Fig. 6g, h). To validate whether *βFaar* functions as ceRNAs to modulate the derepression of *miR-138-5p* targets and impose an additional level of post-transcriptional regulation, target genes of *miR-138-5p* were predicted using TargetScan, starBase, microRNA.org, and miRWalk resources. Major islet specific transcription factors, *Ins2*, *Creb1*, and *NeuroD1*, were identified at the intersection of predictions (Fig. S6e). The predicted binding sites of *miR-138-5p* to *Ins2*, *Creb1* and *NeuroD1* are illustrated in Fig. S6f. Luciferase reporter assay was then applied, which employed pmir-PGLO vector containing wild-type or mutant 3'-UTR of *Ins2*, *Creb1* and *NeuroD1*. We found that the overexpression of *miR-138-5p* reduced the luciferase activity of the WT reporter vector but not mutant reporter vector (Fig. 6i–k). The restraint effect of *miR-138-5p* mimics on *Ins2*, *Creb1* and *NeuroD1* was reversed by the addition of *βFaar* in vitro and ex vivo (Fig. 6l, m; Fig. S6g, h). However, apoptosis induced by the downregulation of *βFaar* could not be reversed by the inhibition of *miR-138-5p* (Fig. S6i). Altogether, these results suggest that *βFaar* regulates β-cell function by competitively binding *miR-138-5p*.

**Decreased *βFaar* increases TRAF3IP2 to activate NF-κB signaling leading to β-cell apoptosis.** *βFaar* is transcribed from the antisense strand in the opposite direction relative to *Traf3ip2* (tumor necrosis factor receptor-associated factor 3 interacting protein 2) and overlaps with the first intron (Fig. S7a). TRAF3IP2 (also known as CIKS or Act1) mediates NF-κB activation through regulating the IκB kinase (IKK)[26,27], and β-cell-specific activation of NF-κB is a key event in the progressive loss of β cells in diabetes[28,29]. This interconnection prompted us to investigate the relationship between *βFaar* and TRAF3IP2. We found that the protein level of TRAF3IP2 was increased in *βFaar* knockdown MIN6 cells and decreased in *βFaar* overexpressing MIN6 cells (Fig. 7a), but had no effect on its mRNA (Fig. S7b). Additionally, TRAF3IP2 protein level was increased in the islets of *βFaar-KO* mice and decreased in the islets of len-*βFaar* mice (Fig. 7b, c). Moreover, TRAF3IP2 was increased in the islets of HFD-fed and db/db mice ($n = 5$) (Fig. 7d, e), and in the islets incubated with 0.5 mmol/l palmitate and pro-inflammatory cytokines (Fig. 7f, g), indicating a negative correlation between *βFaar* and TRAF3IP2 under conditions of obesity. Overexpression of *Traf3ip2* cinduced β-cell apoptosis in vitro and in vivo, while upregulation of *βFaar* abrogated this effect (Fig. 7h, i). Of note, the induction of apoptosis of β-cells by overexpressed *Traf3ip2* was accompanied by phosphorylation of p65 and IκB, and activation of some of the major targets of NF-κB signaling, such as MnSOD, Fas, and iNOS (Fig. 7j). We also found that NF-κB signaling was stimulated in

MIN6 cells transfected with si-*βFaar* and in the islets of *βFaar-KO* mice, and the opposite effect was observed with *βFaar* overexpression (Fig. S7c–e). These results imply that the decreased expression of *βFaar* increases the level of TRAF3IP2, activating the NF-κB signaling and triggering β-cell apoptosis.

To investigate the possible mechanism by which *βFaar* inhibits TRAF3IP2, we identified *βFaar*-interacting proteins by the RNA pull-down assay followed by Coomassie brilliant blue staining and mass spectrometry (MS). Two evident band was subjected to coomassie brilliant blue staining and MS, which highlighted TRAF3IP2 and SMAD specific E3 ubiquitin protein ligase 1 (SMURF1), as potent *βFaar* interacting proteins (Fig. S7g, h). In addition, SMURF1 had higher spectral counts for *βFaar* (Table S2), and it was predicted with higher confidence to be the primary E3 ligase for TRAF3IP2 based on the UbiBrowser resource (http://ubibrowser.ncpsb.org/ubibrowser) (Fig. S7i). Moreover, TRAF3IP2 and SMURF1 were detected in the biotin-labeled sense *βFaar* group by the RNA pull-down assay followed by western blotting (WB) (Fig. 7k). The interaction of *βFaar* with TRAF3IP2 and SMURF1 was further validated by RNA immunoprecipitation (RIP) assay in MIN6 cells. The TRAF3IP2 and SMURF1 antibodies, but not the IgG, successfully pulled down *βFaar* as detected by PCR (Fig. S7j) and qRT-PCR (Fig. 7l). Additionally, a series of truncated *βFaar* constructs were prepared to map the specific binding region between *βFaar* and TRAF3IP2/SMURF1. We found that the 3′ end fragment of *βFaar* (nt 801–1248) was sufficient to bind TRAF3IP2 and SMURF1 (Fig. 7m). Furthermore, *βFaar* colocalized with TRAF3IP2 and SMURF1 in the cytoplasm by FISH/IF (Fig. 7n). All these results confirmed the interaction of TRAF3IP2 and SMURF1 with *βFaar*. SMURF1, whose protein level was no significantly changed in the islets of HFD and db/db mice (Fig. S7k, l), has the activity of ubiquitin E3 ligase in its HECT domain that mediates the ubiquitination of other proteins[30]. Moreover, the endogenous Co-IP assay showed that SMURF1 was an associated protein of TRAF3IP2 (Fig. S7m, n). Interestingly, the interaction of TRAF3IP2 and SMURF1, assessed by Co-IP followed by WB analysis, was facilitated by the overexpression of *βFaar* in vitro and in vivo (Fig. 7o, p). Finally, the overexpression or knockdown *βFaar* did not modulate SMURF1 expression (Fig. S7o-q), and the overexpression of *Smurf1* decreased TRAF3IP2 protein level (Fig. S7r). The identification of these interactions allowed us to hypothesize that *βFaar* functions in protein ubiquitination and protein degradation.

The decrease in TRAF3IP2 upon overexpression of *βFaar* was prevented by the proteasome inhibitor MG132, supporting the role of the ubiquitin/proteasome pathway in controlling the level of TRAF3IP2 (Fig. 7q). We then compared the stability of TRAF3IP2 in MIN6 cells expressing different levels of *βFaar*. Overexpression of *βFaar* rendered TRAF3IP2 highly labile, as measured by assessing the reduction in protein signals following treatment with the translation inhibitor cycloheximide (CHX) (Fig. 7r). Moreover, the knockdown of *βFaar* in MIN6 cells significantly increased the stability of endogenous TRAF3IP2 (Fig. 7s). Next, we determined whether *βFaar* affects the ubiquitination of associated TRAF3IP2. Immunoprecipitation of TRAF3IP2 followed by ubiquitin detection by Western blotting revealed that the overexpression of *βFaar* increased the pool of ubiquitinated TRAF3IP2 in vitro and in vivo, identified as high-molecular-weight discrete or diffuse bands (Fig. 7t, u). Additionally, TRAF3IP2 ubiquitination was inhibited after silencing SMURF1, even in the presence of elevated *βFaar* levels (Fig. 7v, w). Taken together, these data demonstrate that *βFaar* serves as an assembly scaffold that facilitates the interaction of E3 ligases (SMURF1) with the ubiquitination target TRAF3IP2.

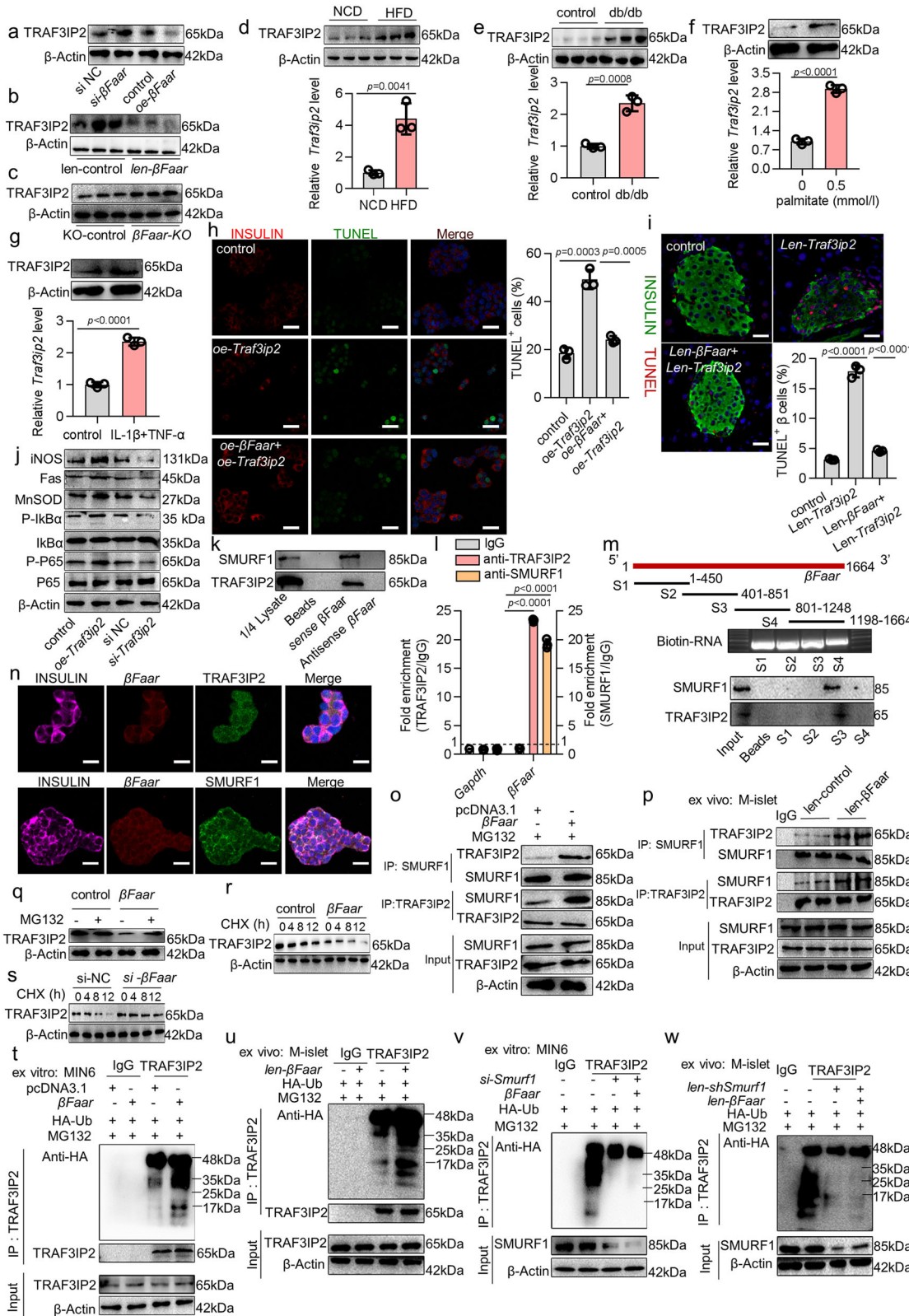

## Discussion

Obesity has detrimental effects on β-cell, resulting in the reduction of insulin content, diminished capacity to secrete insulin in response to glucose, and increased β-cell apoptosis[4]. Despite a major effort to characterize the transcriptome of the different islet cell types to understand the molecular basis of obesity-associated diabetes, very little attention has been paid to the role of lncRNAs

and their contribution to this disease[31,32]. In this study, we have reported that lncRNA *E130307A14Rik*, which we named as the β-cell function and apoptosis regulator (*βFaar*), a 1664 nt antisense non-coding RNA located at chromosome 10: 39621410-39732007, is downregulated in the islets of diabetic db/db mice. The downregulation of *βFaar* is affected by DNMT3a and DNMT3b. Overexpression of *βFaar* protects against HFD-induced β-cell

**Fig. 7 Decreased βFaar increases TRAF3IP2 to activate NF-κB signaling leading to β-cell apoptosis.** The protein level of TRAF3IP2 in the MIN6 cells (**a**), in the islets of len-βFaar mice (**b**) and βFaar-KO mice (**c**, n = 5). The levels of TRAF3IP2 in the islets of HFD mice (**d**, n = 5) and db/db mice (**e**, n = 5). **f**, **g** The levels of TRAF3IP2 in the MIN6 cells. INSULIN⁺/TUNEL⁺ cells in MIN6 cells (**h**) and in the islet of len-βFaar and len-Traf3ip2 treated mice (**i**, n = 3). Scale bars: 10 μm or 20 μm. **j** The protein levels of some major NF-κB signaling targets. **k** TRAF3IP2 and SMURF1 interacted with βFaar. **l** Precipitated RNAs levels in the MIN6 cell lysate was subjected to the anti-TRAF3IP2 or anti-SMURF1 RNA immunoprecipitation (RIP). **m** Mapping of TRAF3IP2/SMURF1-binding domains of βFaar. As shown: schematic diagram of βFaar full-length and truncated fragments (top); gel electrophoresis of in vitro transcribed biotin-labeled RNA of truncated βFaar (middle); western blotting of TRAF3IP2 and SMURF1 in RNA pulldown samples by different βFaar fragments (bottom). **n** Colocalization of βFaar (red), TRAF3IP2 or SMURF1 (green) and INSULIN (pink). Scale bar, 20 μm. **o**, **p** βFaar promoted SMURF1 binding to TRAF3IP2. Cell lysates from MIN6 cells (**o**) or islets from len-βFaar mice (**p**, n = 5). **q** TRAF3IP2 level in MIN6 cells transfected with βFaar. **r**, **s** TRAF3IP2 levels in the MIN6 cells transfected with oe-βFaar (**r**) or si-βFaar (**s**). **t**, **u** βFaar promotes TRAF3IP2 ubiquitination. MIN6 cells were transfected with oe-βFaar (**t**) or primary islets isolated from len-βFaar mice (**u**, n = 5). **v**, **w** SMURF1 promotes TRAF3IP2 ubiquitination. MIN6 cells were transfected with si-Smurf1 or both transfected with si-Smurf1 and oe-βFaar for 48 h (**v**), or islets isolated from len-shSmurf1 and len-βFaar and len-shSmurf1 mice (**w**, n = 5). All experiments above were performed in triplicates, and each group contained three batches of individual samples. The p-values by two-tailed unpaired Student's t test (**d–g**), one-way ANOVA (**h**, **i**), or two-way ANOVA (**l**) are indicated. Data represent the mean ± SD. Source data are provided as a Source data file.

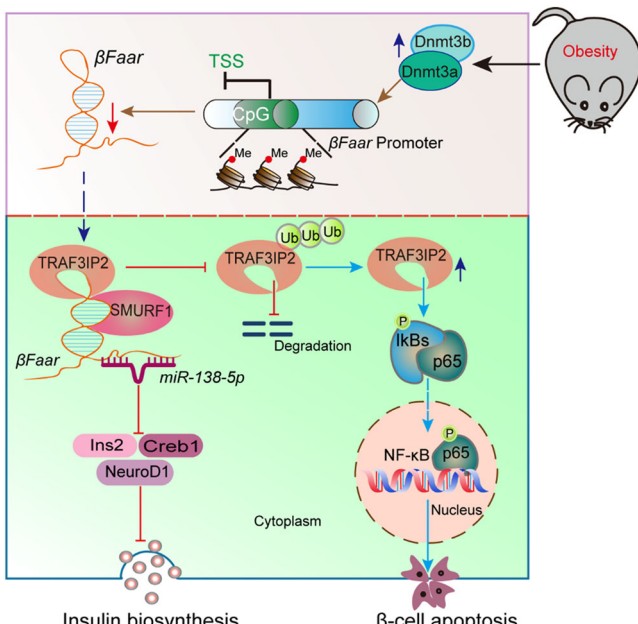

**Fig. 8 Schematic illustration for the mechanism of obesity-induced reduction of βFaar impairing insulin biosynthesis and exacerbated β-cells apoptosis.** During obesity, βFaar is downregulated by DNA hypermethylation, suppressing insulin biosynthesis and secretion by downregulating islet-specific gene (Ins2, NeuroD1, and Creb1) expression through sponging miR-138-5p. Moreover, reduced βFaar inhibits to recruit E3 ubiquitin ligase SMURF1, then suppressing the ubiquitination and degradation of TRAF3IP2, activing the NF-κB signal pathway, and finally inducing β-cell apoptosis.

dysfunction and apoptosis, implying that this lncRNA plays a critical role in the development of obesity-associated diabetes (Fig. 8).

βFaar is transcribed from the antisense strand in the opposite orientation relative to Traf3ip2. Genome-wide association studies (GWAS) demonstrated that human pancreatic islet cells express TRAF3IP2 and that TRAF3IP2 may be involved in the pathogenesis of type 1 diabetes[33]. Moreover, the adapter molecule TRAF3IP2 (also known as CIKS or Act1) was identified as the activator of NF-κB[26,27], while β-cell-specific activation of NF-κB is a key event in the progressive loss of β-cell in diabetes[28,29], raising the intriguing possibility that TRAF3IP2 may be involved in the pathogenesis of β-cell apoptosis. In the present study, we found that obesity affected the expression patterns of βFaar and TRAF3IP2 in an opposite manner. The expression of βFaar decreased in the islets of diabetic db/db and HFD mice, while the expression of TRAF3IP2 increased.

Mechanistically, we first documented that βFaar is colocalized with TRAF3IP2 and SMURF1 in the cytoplasm and then showed the evidence that βFaar forms complexes with E3 ubiquitin ligases SMURF1 and interacts with its ubiquitination substrate TRAF3IP2. By facilitating the formation of these complexes, βFaar promotes the SMURF1-mediated ubiquitination of TRAF3IP2, increasing its degradation. These results uncover the role of a lncRNA, βFaar, as a platform for protein ubiquitination. Furthermore, our results adequately explain the mechanism by which the downregulation of βFaar induces β-cell apoptosis in obesity. The increase of TRAF3IP2 can activate NF-κB pathway[26,27], but what triggers TRAF3IP2 upregulation remains unknown. Our study reveals that βFaar serves as a scaffold facilitating the interaction between the E3 ligase SMURF1 and the ubiquitination target TRAF3IP2, promoting TRAF3IP2 degradation. These findings significantly broaden our understanding of TRAF3IP2/NF-κB pathways.

TRAF3IP2 lies upstream of IKK and JNK, and, in addition to stimulating the classic IKK/NF-κB pathway, it also activates the JNK signaling[26,27]. Interestingly, the JNK pathway regulates various physiological processes, including inflammatory responses, cell death, cell survival, and protein expression[34]. Moreover, JNK in macrophages is required for the development of obesity-induced insulin resistance and inflammation[35]. Therefore, whether βFaar inhibits TRAF3IP2 expression via the JNK signaling pathway requires further investigation. Additionally, we must point out that gene editing (CRISPR/Cas9) or the ectopic expression of βFaar via transgenic technologies were not performed in this study. However, pancreatic ductal infusion of lentivirus vectors was validated as an efficient tool for the ectopic expression of various transcripts[36]. Importantly, we did find out that βFaar is regulated exclusively in the pancreas but not in other tissues, suggesting that pancreatic ductal injection can induce ectopic expression of βFaar specifically in this organ.

Obesity and accompanying insulin resistance may affect DNA methylation[37]. Accumulating evidence has indicated that DNA methylation would serve as a bridge between environmental changes and cellular responses. Of note, nutrient status differentially modulates DNA methylation in several metabolic genes including hepatocyte nuclear factor 4α (HNF4α)[38], pancreatic and duodenal homeobox 1 (PDX1)[39], and peroxisome proliferator-activated receptor gamma coactivator-1α (PGC-1α)[40]. Intriguingly, several reports have demonstrated that HFD consumption affects the epigenetics of various genes involved in inheriting metabolic imbalance by the offspring[41]. Mechanistically, the change in methylation level caused by obesity is mediated by different types of DNMTs. Obesity-induced pro-inflammatory cytokines promote the expression and enzymatic activity of DNMT1 in adipose tissue and acute myeloid leukemia (AML) cells[42,43]. In our study, we documented a

significant increase in DMNT3a/b, but not DNMT1, in the islets of diabetic db/db mice, a result consistent with the findings of Dhawan and coworkers[44]. This change suggests that obesity may cause the ectopic expression of distinct kinds of DNMTs in different tissues, highlighting the complexity of the regulation of gene expression by obesity.

Given their broad definition, lncRNAs represent a heterogeneous group of transcripts, a subset of which modulates gene expression through diverse mechanisms. lncRNAs have been proposed to act as scaffolds for chromatin modifiers, blockers of transcription, antisense RNAs, microRNA sponges, protein decoys, and enhancers[45]. Our data showed that, in addition to acting as a miR-138a-5p sponge, βFaar serves as a molecular scaffold to facilitate the interaction between the E3 ligase SMURF1 and its ubiquitination target TRAF3IP2. Although the lncRNA βFAAR (TRAF3IP2-AS1) has been reported to be associated with tumors[46], and upregulated in cocaine abusers[47]. Interestingly, the expression of human lncRNA βFAAR (TRAF3IP2-AS1) was predicted in human β cells by Akerman[48]. In our study, we explore the role of mus-βFaar and has-βFAAR in regulation of β-cell function, our results showed that overexpression of βFaar can improve β-cell function.

In conclusion, our investigation revealed an important role of βFaar in the development of obesity-associated β-cell dysfunction and apoptosis. βFaar promotes insulin synthesis and secretion by upregulating the expression of Ins2, NeuroD1, and Creb1 through sponging miR-138-5p and by acting as a scaffold function to assemble E3 ubiquitin ligases and its ubiquitination substrates. Also, βFaar serves as a platform to control protein levels via the ubiquitin-proteasome pathway. Therefore, our present study sets the stage for the development of βFaar, TRAF3IP2, and potentially other βFaar interacting molecules as therapeutic targets for obesity-associated diabetes and other obesity-associated diseases.

## Methods

**Animal care**. Care of all animals was within institutional animal-care committee guidelines, and all procedures were approved by the animal ethics committee of China Pharmaceutical University (Permit Number: 2162326) and were in accordance with the international laws and policies (EEC Council Directive 86/609, 1987). All animals were on the C57BL/6 background except for db/− mice and db/db mice, which were on the BKS background. Mice were housed in a pathogen-free animal facility and maintained in a temperature-controlled room (22 °C), with humidity at 55% and on a 12 h light-dark cycle (lights on from 6 a.m. to 6 p.m.). Unless otherwise stated, animals were fed normal chow diet (D12450J, 10% calories from fat), and water ad libitum. Diet-induced obesity was obtained by feeding a high-fat diet (D12494, 60% calories from fat) for at least 8 weeks. Mice then were sacrificed under isoflurane inhalation (1.4%) followed by cervical dislocation. The age of mice is either indicated in the Figures or was above 8 weeks of age. Male mice were used for all studies shown.

**Isolation and culture of primary islet cells**. Human islets were provided from Tianjin First Central Hospital. All human studies were conducted according to the principles of the Declaration of Helsinki and approved by Ethics Committee of the Tianjin First Central Hospital[49]. Written informed consent was obtained from all subjects. High purity islets (>80%) were collected and cultured in CMRL-1066 medium (Corning, Manassas, VA, USA), supplemented with 10% Human Serum Albumin (Baxter, Vienna, Austria), 100 U/mL penicillin and 100 μg/mL streptomycin at 37 °C in 5% CO₂.

Mice islets were isolated by collagenase digestion and enriched using a Histopaque (Sigma Aldrich) density gradient[50]. Isolated islets were collected and resuspended in RPMI-1640 medium (glucose: 11.1 mmol/l) containing 10% FBS, 100 IU/ml penicillin, and 100 μg/ml streptomycin at 37 °C in a humidified 5% CO₂ atmosphere. After equilibrating for 3 h, the islets were counted and replanted to 6- or 48-well plates and cultured overnight for further experiments. For pathophysiological concentrations of palmitate, glucose, and proinflammatory cytokines treatment, islets were incubated in modified medium with 0.5% (weight for volume) BSA, various concentrations of glucose (low glucose: 2.5 mmol/l; high glucose: 33.3 mmol/l), palmitate (0.5 mmol/l), IL-1β + TNFα (IL-1β: 5 ng/ml; TNFα: 30 ng/ml).

**Cell culture**. The mice pancreatic β cell line MIN6 was donated by Defu Zeng, Professor, from Departments of Diabetes Immunology and Hematopoietic Cell Transplantation Irell & Manella Graduate School of Biological Sciences of City of Hope. And MIN6 cells were maintained in DMEM (Gibco) containing 15% FBS (Gibco, Burlinton, ON, USA), 100 IU/mL penicillin, 100 μg/ml streptomycin, and 50 μmol/l β-mercaptoethanol (Sigma Aldrich, St. Louis, MO, USA) at 37 °C in humidified atmosphere containing 5% CO₂. For pathophysiological concentrations of palmitate, glucose, and proinflammatory cytokines treatment, MIN6 cells were incubated in modified medium with 0.5% (weight for volume) BSA, various concentrations of glucose (low glucose: 2.5 mmol/l; high glucose: 33.3 mmol/l), palmitate (0.5 mmol/l), IL-1β + TNFα (IL-1β: 5 ng/ml; TNFα: 30 ng/ml).

**Insulin secretion assay**. MIN6 cells ($2 \times 10^5$ cells well⁻¹) or isolated mouse islets (15 islets well⁻¹) were seeded in 48-well plates and ectopically expressed βFaar as above for 48 h for glucose-stimulated insulin secretion (GSIS) assay. MIN6 cells or the islets were pre-incubated overnight in KRBH balanced buffer containing 0.2% BSA supplemented with 2.5 mmol/l glucose, and were incubated for 2 h in 2.5 mmol/l or 33.3 mmol/l glucose. Immediately after incubation an aliquot of the medium was removed for analysis of insulin, and the cells were incubated in acid-ethanol for insulin content determination by mice insulin ELISA kit (ExCell Bio, Shanghai, China), according to the manufacturer's instructions.

Human pancreatic islets were overexpressed or knockdown of βFAAR for 48 h (12 islets well⁻¹). Thereafter the islets were washed and preincubated for 30 min at 37 °C in Krebs Ringer bicarbonate buffer (KRB), pH 7.4, supplemented with HEPES (10 mM), 0.1% bovine serum albumin, and 1 mmol/l glucose. After preincubation, the buffer was changed to a medium containing either 1 mmol/l or 16.7 mmol/l glucose. The islets were then incubated for 1 h at 37 °C. Immediately after incubation an aliquot of the medium was removed for analysis of insulin, and the islets were incubated in acid-ethanol for insulin content determination by human insulin ELISA kit (ExCell Bio, Shanghai, China), according to the manufacturer's instructions.

**RNA-sequencing analysis**. Total RNA from islets (more than 100 islets per group) of wide type control mice (n = 3) and db/db mice (n = 3) were isolated using the RNeasy mini kit (Qiagen) following the protocol. The quality of the samples, the experiment, and the analysis data was completely finished by the Vazyme (Nanjing, China). Cuffdiff (v2.2.1)[51] was used to calculate the fragments per kilobase million (FPKM) for lncRNAs in each group. A difference in gene expression with a p value ≤ 0.05 was considered significant. The raw data are displayed in Table S3.

**Subcellular fractionation**. MIN6 cells ($10^6$ cells) were incubated for 3 min and primary islets (100 islets) were incubated for 5 min in ice-cold Cell Fractionation Buffer (PARIS™ Kit, life), and then centrifuged for 5 min at $500 \times g$. The supernatant (cytoplasmic fraction) was recovered, while the pellet was resuspended in ice-cold Cell Disruption Buffer for 10 min at 4 °C. The samples were then centrifuged at $500 \times g$ for 5 min, and the pellet was collected as the nucleoplasmic fraction.

**Fluorescence in situ hybridization (FISH)**. Cy3 labeled βFaar probe was designed and synthesized by GenePharma (Shanghai, China). For FISH assay, MIN6 cells ($1 \times 10^5$ cells) or primary islets (10 islets) were fixed in 4% formaldehyde and permeabilized with 0.3% Triton X-100 for 15 min, washed with PBS three times and once in 2× SSC buffer. Hybridization was carried out using DNA probe sets at 37 °C for 16 h. Images were obtained with confocal laser scanning microscope (CLSM, LSM700, Zeiss, Germany) and processed using the ZEN imaging software.

**Plasmid construction**. The coding sequences for TRAF3 Interacting Protein 2 (Traf3ip2) (NM_134000.3), SMAD Specific E3 Ubiquitin Protein Ligase 1 (Smurf1) (NM_001038627.1) and the sequence of βFaar (NR_038037.1) were amplified by PCR from full-length cDNA of mice, and then cloned in pcDNA 3.1 vector (Addgene, Watertown, MA, USA). The mouse βFaar, Dnmt1, Dnmt3a and Dnmt3b promoter (the sequence was obtained from UCSC) were obtained by PCR. Then the PCR product was cloned into the PGL3-basic vector (Addgene, Watertown, MA, USA). All plasmids were confirmed to be correct by sequencing. To construct the reporter plasmids, the complete 3'-UTR of murine βFaar, NeuroD1 (NM_010894.3), Creb1 (NM_009952.2) and Insulin2 (NM_001185084.2), containing either wild type or mutated binding sites of miR-138-5p, was inserted behind the firefly-luciferase gene of pmir-PGLO vector (Addgene, Watertown, MA, USA) and was located between the Xhol I and Xbal I restriction sites. All the primer sequences for PCR are listed in Table S4.

**Plasmid and transient transfections**. MIN6 cells (~$5 \times 10^5$) were seeded in six-well plates in culture medium without antibiotics and transfected with full-length cDNA encoding βFaar, Traf3ip2, and Smurf1 or pcDNA 3.1 plasmid (non-coding) using Lipofectamine 2000 reagent (Invitrogen) according to the manufacturer's instructions. After 48 h transfection, the cells were harvested and analyzed by western blot and qRT-PCR for the relative level of various proteins and mRNA.

miR-138-5p duplex mimics (50 nmol/l), 2′-O-methylated single-stranded miR-138-5p antisense oligonucleotides (anti-miR-138-5p, 100 nmol/l), and negative controls at the same concentration were obtained from GenePharma (Shanghai, China). For transient transfection, isolated mouse islets (100 islets) or MIN6 cells

(~$5 \times 10^5$) were seeded in six-well plates cultured in media without antibiotics. Lipofectamine 2000 reagent (Invitrogen) was mixed with *miR-138-5p* mimics/inhibitors according to the manufacturer's instruction, mimics NC (miR-NC) or inhibit NC (anti-NC) was transfected at the same concentration as negative control. At 48 h post-transfection, the cells were harvested and analyzed by qRT-PCR for the relative level of *miR-138-5p*.

**Luciferase assays.** MIN6 cells were plated at the concentration of $2 \times 10^5$ cells well$^{-1}$ in 24-well plates for 24 h. 0.9 µg DNA well$^{-1}$ (0.4 µg construct promoter, 0.4 µg transcription factors and 0.1 µg constitutive renilla expression plasmid as a control for transfection efficiency) was transfected into MIN6 cells for 24 h. Then, luciferase activities were measured using a dual-luciferase reporter assay system (Vazyme, Nanjing, China). For the generation of reporter constructs, the complete 3'-UTR of murine *βFaar*, *NeuroD1*, *Creb1,* and *Ins2* containing either the wild type or mutated *miR-138-5p* binding sites was cloned behind the stop codon of the firefly-luciferase open reading frame using specific primers. 100 ng pmir-PGLO reporters along with *miR-138-5p* mimic or *miR-138-5p* inhibitor were transfected into MIN6 cells by using Lipofectamine 2000 transfection reagent (Invitrogen). At 24 h post-transfection, dual-luciferase reporter assays were performed using a Luciferase Assay System (Vazyme, Nanjing, China). Transfection data represent at least three independent experiments each performed in triplicates. The wide type and mutation site primers are listed in Table S4.

**Bisulphite sequencing.** Genomic DNA was extracted from MIN6 cells treatment with 0.5 mmol/l palmitate or extracted from the islets of db/db or control mice (n = 7). The DNA was subjected to sodium bisulphite modification using EpiTect Bisulfite Kits (QIAGEN) following the manufacturer's protocols. Sodium bisulphite-treated DNA was PCR amplified using primers (Table S4) predicted on MethPrimer website and EpiTaq™HS (Takara). Amplified bisulfate PCR products in each group were cloned into the T-Vector pMD19 (Takara), and over 20 clones were sequenced.

**Transfection of lncRNA smart silencer.** LncRNA smart silencer synthesized from RiboBio (Guangzhou, China), which is a mixture of three siRNAs and three antisense oligonucleotides (ASOs) for knockdown βFaar in cytoplasm and nucleus, respectively. The target sequences were listed in Table S5. The smart silence negative control (si-NC) does not contain domains homologous to human, mice and rats. For transient transfection, isolated mouse islets (100 islets) or MIN6 cells (~$5 \times 10^5$) were seeded in six-well plates, cultured in media without antibiotics and transfected with Lipofectamine 2000 reagent (Invitrogen) according to the manufacturer's instructions. Cells were transfected for 24 h with the βFaar silencer at a final concentration of 50 nmol/l or with control si-NC at the same concentration before changing to fresh media including antibiotics. At 48 h after transfection, cells were lysed to extract total RNA to measure the knockdown efficacy.

**Flow cytometric analysis of apoptosis.** MIN6 cells ($1 \times 10^6$ cells/well) were cultured in 6-well plates and ectopically expressed βFaar or Traf3ip2 for 48 h. The cells were harvested and treated with EDTA for 5 min, according to the instructions of Annexin V-FITC Apoptosis detection kit (Vazyme, Nanjing, China). After the double staining with FITC-Annexin V and Propidium iodide (PI), the cells were analyzed with a flow cytometry (FACScan®; BD Biosciences) equipped with FlowJo v10 software (BD Biosciences).

**Lentiviral production.** βFaar or Traf3ip2 overexpression plasmid was constructed in the pHAGE-CMV-Rat Insulin2 promoter (pHAGE-CMV-RIP) vector using the One Step Cloning Kit (C113-02, Vazyme Biotech Ltd., Nanjing, China). βFaar-sgRNA was constructed in lentiCRISPRv2-Rat Insulin 2 promoter (lentiCRISPRv2-RIP) vector using BsmB I. Lentiviral particles were produced by transient transfection of HEK293T cells with the packaging plasmids pMD2.G (Addgene #12259) and psPAX2 (Addgene #12260), together with the lentiviral βFaar- or Traf3ip2-overexpression vector (len-βFaar, len-Traf3ip2) using Lipofectamine 3000 (Invitrogen, Carlsbad, CA, USA). Lentiviral particles carrying βFaar-sgRNA-expressing vectors (βFaar-KO) were also obtained from HEK293T cells using the same method. The culture medium was collected 48 h after transfection of HEK293T, and lentiviral particles were concentrated by ultracentrifugation. For each construct, the lentiviral titer was determined by infection of HEK293T cells and FACS analysis.

**Pancreatic intra-ductal lentivirus infusion in mice.** The procedure for retrograde infusion of lentivirus into pancreatic duct has been described by Xiao et al[36]. Briefly, the male C57BL/6J mice (8 weeks old, about 23–25 g) were anesthetized with isoflurane. A midline incision was made to reveal the abdominal cavity. Q-tips were used to gently pull out the stomach. Rotate and stretch the duodenum to expose the biliary-pancreatic duct and its junction with the duodenum. A microclamp was placed on the bile duct above the branching of the pancreatic duct to prevent perfusion of the liver. Then a 30-gauge needle was inserted through the anti-mesenteric aspect of the duodenum to cannulate the common bile duct and the tip of the catheter should be positioned at the origin of the pancreatic duct

branch in the biliary-pancreatic duct. A small microclamp was applied to the distal common bile duct to prevent back flow of the infusate into the duodenal lumen and to hold the cannula in place. $1 \times 10^9$ indicated lentivirus dissolved in 0.15 ml normal saline (NS) or normal saline (NS) was infused at 6 µl per min for 25 min using a R462 perfusion pump (RWD, China, shengzhen). After infusion, close the hole in the duodenum created by the catheter with a 7-0 suture. Upon completion of the infusion, the clamps were released. The exterior abdominal wound was closed using 7 mm wound clips. Mice recovered on a heating pad (37 °C) until it fully recovers. They were given free access to food and water after the surgery. At 72 h after injection, islets were lysed to extract total RNA or protein to measure the overexpression efficacy. At 1 week after injection, mice were fed with high fat diet (HFD, D12494, 60% energy from fat) for 16 weeks.

**Mouse metabolic assays.** After 12 h fasting treatment, mice fasting blood glucose (FBG) levels and fasting serum insulin (FINS) levels were examined via using a glucometer (OMRON, Japan) and by ELISA (ExCell Bio, Shanghai, China), respectively. And the homeostatic model assessment indices of insulin resistance (HOMA-IR) was calculated with the equation (FBG (mmol/l) × FINS (mIU/l))/22.5. To perform the glucose tolerance tests, 2 g/kg glucose (Sigma-Aldrich, St Louis, MO, USA) was intraperitoneal (i.p.) injected into mice, whereas 0.75 U/kg insulin (Novolin R, Novo Nordisk, Bagsvaerd, Denmark) was i.p. injected into mice for insulin tolerance tests. Blood glucose levels were examined at 0, 15, 30, 60, 90, and 120 min after injection and serum sample was collected from eye canthus blood at 0, 5, 15, and 30 min after glucose injection. Insulin level was evaluated using mice insulin ELISA kit (Crystal Chem, USA), according to the manufacturer's instructions. The AUC, calculated by the conventional trapezoid rule, are given as the incremental area under the curve.

**RNA pull-down.** Firstly, in vitro translation assays were performed using T7 RNA Polymerase transcription kit according to the manufacturer's instruction (Thermo, MA, USA), total RNA treated with RNase-free DNase I (Roche, Basel, Switzerland) and then purified with RNeasy Mini Kit (QIAGEN). Then βFaar RNAs were labeled by using Pierce RNA 3' End Desthiobiotinylation Kit (Thermo). Biotinylated RNAs were incubated with cytoplasmic extract of MIN6 cells ($5 \times 10^6$) at room temperature for 2 h. Washed streptavidin agarose beads (Invitrogen) were added to each binding reaction and incubated at room temperature for 1 h. Recovered proteins associated with βFaar or control were resolved by gel electrophoresis and coomassie brilliant blue. The eluted solutions were accomplished by mass spectrometry analysis (Shanghai Applied Protein Technology Co., Ltd) on Q Exactive mass spectrometer (Proxeon Biosystems, now Thermo Fisher Scientific) or western blot.

For RNA-RNA pull down assays, MIN6 cells ($5 \times 10^6$) were lysed using lysis buffer to obtain cell lysate (Millipore, 17-700). Biotin-labeled *miR-138-5p* and its control RNA (Genepharma, Shanghai, China) were incubated with cell protein extracts, and added with streptavidin magnetic beads and incubated over night at 4 °C. The expression levels of βFaar and *miR-138-5p* were detected by qRT-PCR.

**RNA isolation and qRT-PCR analysis.** Primary islets and MIN6 cells were cultured and treated as described above. Total RNA was extracted using TRIzol (Invitrogen) and an RNeasy kit (QIAGEN, Duesseldorf, Germany)[15]. Reverse transcription reaction was performed using PrimeScriptTM RT reagent Kit (Takara, Tokyo, Japan) and diluted cDNA was used for qRT-PCR analysis using SYBR Premix Ex Taq II Kit (Takara) with the appropriate primers listed in Table S6. Relative expression of genes was determined using a comparative method ($2^{-\triangle\text{CT}}$). U6 and Gapdh were used as internal standards for miRNAs and mRNAs, respectively. For *miR-138-5p* and U6, TaqMan probes (Ambion) were used to confirm our results.

**RNA immunoprecipitation (RIP).** RNA immunoprecipitation was performed using the EZMagna RIP kit (Millipore, Billerica, MA, USA) following the manufacturer's protocol. MIN6 cells were lysed in complete RIP lysis buffer, after which 100 µl of whole-cell extract was incubated with RIP buffer containing magnetic beads conjugated with anti-TRAF3IP2 antibody (Abcam), anti-SMURF1 antibody (Abcam), or anti-Ago2 antibody (CST), negative control normal mouse IgG (Abcam). Samples were incubated with Proteinase K with shaking to digest the protein and then immunoprecipitated RNA was isolated. The RNA concentration was measured using a Microplate reader (Synergy2, BioTek, USA) and the RNA quality assessed using a bioanalyser (Agilent, Santa Clara, CA, USA). Furthermore, purified RNA was subjected to qRT-PCR analysis to demonstrate the presence of the binding targets using respective primers. The primer sequences were listed in Table S6.

**Immunohistochemistry and immunofluorescence.** Pancreas and white adipose were fixed in 4% paraformaldehyde and embedded in paraffin, and the antigen from the cut sections were retrieved by boiling them in 10 mmol/l Tris/EDTA (pH 9.0). Sections were permeabilized and blocked in PBS buffer containing 0.3% Triton X-100, 1% BSA, and 5% goat serum. Primary antibody binding was performed overnight at 4 °C, while incubation with secondary antibody was done at room temperature for 1 h. The slides were analyzed using a confocal laser scanning

microscope (CLSM, Carl Zeiss LSM700) at ×20 or ×40 magnification. The antibodies are listed in Table S7.

**Western blot analysis**. The mouse islets and MIN6 cells were cultured and treated as described above and lysed with RIPA Lysis Buffer (Beyotime) containing 1% PMSF (Sigma). After protein content determination, Western blot analyses were conducted according to standard procedures using specific antibodies. The antibodies are listed in Table S7. Densitometric calculations were expressed as fold change in proteins relative to β-Actin expression levels by ImageJ software.

**Insulin/TUNEL double staining**. MIN6 cells and primary islets were fixed in 4% paraformaldehyde, permeabilized and blocked in PBS buffer containing 0.3% Triton X-100, 1% BSA, and 5% goat serum. Primary INSULIN antibody binding was performed overnight at 4 °C, while incubation with secondary antibody was done at room temperature for 1 h. Then, TUNEL reaction mixture was incubated 1 h at 37 °C using a TdT-dUTP nick-end labeling (TUNEL) apoptosis assay kit (Beyotime, China, shanghai). Finally, cells were washed and nuclei were stained with DAPI. Images were obtained using a confocal laser scanning microscope (CLSM, Carl Zeiss LSM700) at ×40 or ×20 magnification.

**Morphometry**. For determining β cell mass, pancreatic sections were stained with anti-insulin antibodies and DAPI, the pancreatic sections were scanned entirely using a ×10 objective of a Zeiss LSM700 microscope. The fraction of the insulin-positive areas were determined using Image J, and the mass was calculated by multiplying this fraction by the initial pancreatic wet weight.

The number of β-cells/pancreas area was determined from counts of β-cells (DAPI$^+$ nuclei in insulin$^+$ area) on sections of 10–20 islets/mouse[52].

**Cell counting kit-8 (CCK-8) assay**. MIN6 cells were seeded in 96-well plates (4 × $10^4$ cells well$^{-1}$) in 100 μl culture medium. CCK-8 assay (Vazyme, Nanjing, Jiangsu, China) was performed at 0, 24, 48, and 72 h after *oe-βFaar* or *si-βFaar* transfection, according to the manufacturer's instructions.

**Statistical analysis**. Data are presented as mean ± SD or mean ± SEM. Comparisons were performed using the Student's *t* test between two groups or ANOVA in multiple groups. Dunn's multiple comparisons for one-way ANOVA and Fisher's least significant difference (LSD) for two-way ANOVA were used. The level of significance was set at *$p < 0.05$, **$p < 0.01$, ***$p < 0.001$. Graphpad prism 7 (GraphPad, San Diego, CA, USA) was used for all calculation.

**Reporting summary**. Further information on research design is available in the Nature Research Reporting Summary linked to this article.

## Data availability

The authors declare that all data supporting the findings of this study are available within the article and its Supplementary Information files. The raw data for dot graphs and uncropped versions of any gels or blots or micrographs presented in the figures and Supplementary figures are included in the Source data file. The RNA-seq raw data that support the findings of this study has been deposited in the NCBI's Sequence Read Archive (SRA) database (PRJNA681104). The vectors or mouse models used in this study will be available from the corresponding author upon reasonable request. Source data are provided with this paper.

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

## Acknowledgements

This work was supported by National Natural Science Foundation of China (Grant No. 82070801, 81570696, 81570739, and 81970717); supported by grants from the "111" project (B16046); supported by Priority Academic Program Development of Jiangsu Higher Education Institutions (PAPD); supported by China Postdoctoral Science Foundation (2020M671661); supported by Jiangsu Province Science Foundation for Youths (BK20200569); supported by Jiangsu Province Research Founding for Post-doctoral (1412000016).

## Author contributions

F.F.Z., Y.Y., X.C., Y.L., Q.X.H., B.H., Y.H.L., and G.Q.L. performed the experiments; Y.P. and Y.F.Z. analyzed data; D.C.L. and R.L. provided human islets; F.F.Z., W.Q., L.L., and L.J. designed the project, interpreted the data, and wrote the manuscript.

## Competing interests

The authors declare no competing interests.
