## [Peer Review File · Nature Communications]

Editorial Note: Parts of this Peer Review File have been redacted as indicated to maintain the confidentiality of a third party certificate and an invoice document.”

REVIEWER COMMENTS

Reviewer #1 (Remarks to the Author):

Zhang et al

The authors have taken an elegant and logical approach to identifying novel long non-coding RNAs which may be dysregulated in the context of type 2 diabetes, in this case diabetic db/db mouse islets. In an unbiased screen using RNASeq they find that a lnc expressed at Ch10qk B1 in mice (6aq21 in man), which they have renamed bFaar, is down-regulated in the above model. They go on to show that forced over- or under-expression of this lnc is associated with lowered or increased expression of key beta cell genes, including insulin. bFaar expression is decreased by palmitate via altered DNA methylation, resulting from elevated DNA methyl transferase activity (DNMT3a and -3b). Rescue experiments suggest that bFaar acts as a sponge for Mir-138-5p which itself is a negative regulator of insulin gene expression, and also prevents ubiquitination and destruction of TRAF3IP2. The report represents a substantial amount of work and provides an impressive and interesting story.

Major

1. All reported data are from rodent systems. Although I do not feel it essential for the authors to recapitulate their findings in human beta cells, this limitation should be stated explicitly in the Abstract and in the Discussion section
2. Further evidence corroborating an action on NF-KB signalling would reinforce the conclusions of the manuscript, and could include measurement of the subcellular distribution of NF-KB subunits, or phosphorylation of IK-B, after manipulation of bFaar expression.
3. What is the distribution of bFaar expression between beta and other islet cell types?
4. Can the authors shed more light on the intracellular distribution of bFaar?

Minor

1. The method of overexpression or KO in vivo - intraductal infusion; <https://www.ncbi.nlm.nih.gov/pmc/articles/PMC4734891/> affects all pancreatic cells, including exocrine. What was the proportion of islets is affected overall, and how does this impact the interpretation of the data both in isolated islets and in vivo (glucose tolerance)? This may or may not be a problem, but surely it's better to have islet/beta cell-selective expression (insulin promoter/ optimized viral serotype?)
2. Are other DNMT3a, -3b targets affected by palmitate?
3. There are issues throughout with English and grammar that require attention.

Reviewer #2 (Remarks to the Author):

RE: NCOMMS-20-35111-T

The paper described the discovery of β Faar, a lncRNA, that is highly expressed in the mouse pancreatic insulin-secreting β cells. Using both in vitro and in vivo approaches the authors demonstrated that β Faar positively regulates insulin biogenesis and protects β cells from obesity-induced apoptosis. Mechanistically, the authors provided compelling evidence that β Faar stimulates insulin biosynthesis by sponging miR-138-5p whereby derepressing key transcription factors Ins2, Creb1 and NeuroD1 known to regulate insulin transcription. On the other hand, the authors showed that β Faar acted as an assembly scaffold for TRAF3IP2 and SMURF1 to induce ubiquitination and degradation of TRAF3IP2. Specifically, TRAF3IP2 activated NF- κ B signaling to induce β cell apoptosis and β Faar protected β cells by downregulating TRAF3IP2. The paper is nicely written, the mode of action of β Faar is novel and intriguing, and the conclusions are overall well supported by the data.

However, I have a major concern about the conservation of β Faar between human and mouse. A common rule of thumb for two sequences considered to be homologous is that they are more than 30% identical over their entire lengths. As shown in Supplementary Fig. 1c-f, the overall identity between the mouse β Faar and the putative human β Faar is less than 30%, indicating that they are not conserved based on the sequence alone. A careful definition of conservation would require more advanced methods involving statistic models such as PhyloP and PhyloHMM. Numerous lncRNAs identified so far in the mouse have been shown to have no functional counterparts in the human. It is very important that the authors provide evidence on which of the putative human β Faar RNA(s) are actually expressed in human pancreatic β cells and if so, whether they perform functions similar to the mouse β Faar. Human pancreatic β cell lines should be available (Scharfmann et al, The supply of chain of human pancreatic β cell lines. JCI, 2019, 129(9):3511-3520). Otherwise, the statement “characterize the β Faar as a potential target for treatment of obesity-associated diabetes” (lines 35-36) is invalid, and the translational significance of this work is severely diminished.

Minor points:

1. The evidence for DNMT3a and DNMT3b involved in obesity-induced β Faar hypermethylation and downregulation is weak. Increased DNMT3a and DNMT3b expression was only shown in the pancreatic islets from db/db mice. The authors need to show whether this is also the case for other obese models, such as HFD. Otherwise, the authors should remove DNMT3b and DNMT3a from Fig. 8. Without data from human β cells the authors should replace the human cartoon with a mouse.
2. In Fig. S3a, and b, the 450-fold increase in the level of β Faar by oe- β Faar was an exaggeration. I suspect it was likely an artifact from a contamination of the transfected plasmid DNA (i.e., incomplete DNase digestion prior to RT-qPCR). Cells transfected with the β Faar plasmid would retain the plasmid and that a small fraction of it would escape DNase digestion during RNA purification and be amplified by

RT and the highly sensitive PCR. The authors can test this possibility by putting through RNA purification 1/5-1/10 amounts of the β Faar plasmid DNA used to transfect the cells, followed by RT-qPCR. I am sure they will see strong β Faar signals that at least in part actually come from the contaminated plasmid DNA.

3. Lines 140-142: the sentence “We next confirmed that β Faar was re-expressed in MIN6 cells and primary islets after DNA demethylation treatment with 5-Aza-dC in a dose-dependent manner (Figure 2c)” is a bit misleading. It gave one the impression that β Faar expression was first inhibited (by palmitate or cytokines) and then derepressed by 5-Aza-dC. It should be rephrased to something like “Treatment of MIN6 cells and primary islets with 5-Aza-dC increased β Faar expression in a dose-dependent manner”.

4. Line 175: “Camp” should be “cAMP”.

Reviewer #3 (Remarks to the Author):

In this manuscript, the authors identified a lncRNA (named β Faar) dramatically downregulated in the islets of obese mice and necessary for β -cell dysfunction and apoptosis. As to the mechanisms referring to β -cell apoptosis during obesity, the authors showed that β Faar could negatively regulate expression of TRAF3IP2, a NF- κ B pathway activator, and provided a scaffold to boost the interaction between TRAF3IP2 and Smurf1, a known HECT type E3 ligase. This conclusion was highlighted by the authors in the abstract, emphasizing a potential important role of Smurf1 involved in NF- κ B regulation and β -cell dysfunction. It is a new knowledge about Smurf1 function, which is impressive for Smurf1 biology. However, these data mainly derived from cells with genes artificial overexpression or knockdown manipulation, it's preliminary and not convincing. More in-vivo evidence of β Faar-TRAF3IP2-Smurf1 regulation and related β -cell function should be provided.

1. First, the authors need to manipulate β Faar or TRAF3IP2 or Smurf1 in vivo, by the way of overexpressing or knocking down, to recapitulate the conclusion achieved in cells. For example, β Faar should be decreased in mice to see if TRAF3IP2 was increased, to examine if NF- κ B was activated, to test if β -cell apoptosis was observed. Further, β -cell area, or insulin secretion activity should be examined.

2. Also, the authors showed evidence in cells that β Faar forms complexes with Smurf1 and TRAF3IP2, and promotes the interaction and ubiquitination of TRAF3IP2 by Smurf1. I strongly suggest the authors to examine the interplay (interaction and ubiquitination assays) between TRAF3IP2 and Smurf1 in vivo by overexpressing β Faar in mice model.

3. In Fig.7, the authors gave a detailed information about the interplay between β Faar and TRAF3IP2. However, when Smurf1 is introduced into the three-part interaction, how Smurf1 is regulated by β Faar, how Smurf1 expression is changed under HFD treatment or in db/db mice, and how Smurf1 is involved

in NF- κ B pathway are missing. The authors should design a series of experimental data in detail to tell us whether Smurf1 is truly involved in this regulation.

Dear reviewers:

Thank you very much for your comments and advice to our manuscript entitled “**Obesity-inhibited LncRNA *βFaar* regulates islet β -cell function and survival**”. We completely accept your recommendation and fully agree that these recommendation can further strength the quality of the manuscript. We have revised the manuscript very carefully and according to the suggestion. To clearly present the response, the comments are shown in *italics* and our responses are shown in blue font. A thorough, point-by-point response to each point was raised, and a word file of the revised manuscript with all changes labelled in red font has been uploaded. If you have any further questions about the revision, please do not hesitate to contact us.

Best regards,

Liang Jin

Comments:

Reviewer #1 (Remarks to the Author):

Zhang et al

The authors have taken an elegant and logical approach to identifying novel long non-coding RNAs which may be dysregulated in the context of type 2 diabetes, in this case diabetic db/db mouse islets. In an unbiased screen using RNASeq they find that a lnc expressed at Ch10qk B1 in mice (6aq21 in man), which they have renamed bFaar, is down-regulated in the above model. They go on to show that forced over- or under-expression of this lnc is associated with lowered or increased expression of key beta cell genes, including insulin. bFaar expression is decreased by palmitate via altered DNA methylation, resulting from elevated DNA methyl transferase activity (DMNT3a and -3b). Rescue experiments suggest that bFaar acts as a sponge for Mir-138-5p which itself is a negative regulator of insulin gene expression, and also prevents ubiquitination and destruction of TRAF3IP2. The report represents a substantial amount of work and provides an impressive and interesting story.

Response: Thanks for your positive comments. As suggested, we provided more convinced evidences to strengthen our conclusions. For example, we demonstrated that overexpression of *Traf3ip2* induces apoptosis of β -cell accompanied by phosphorylation of P65 and I κ B, and activation of some major NF- κ B signaling targets (MnSOD, Fas, and iNOS) (Figure 7j). We also found that NF- κ B signaling was activated in the MIN6 cells transfected with *si- β Faar* and in the islets of *β Faar-KO* mice, and vice versa (Figure S7 c-e). Immunofluorescence results further revealed that *β Faar* expression was enriched in β cells (Figure S1e). We hope the explanations and changes above would make you and other readers much easier to understand our manuscript.

Major

1. All reported data are from rodent systems. Although I do not feel it essential for the authors to recapitulate their findings in human beta cells, this limitation should be stated explicitly in the Abstract and in the Discussion section.

Response: Thanks for your encouraging comments. As suggested, we have stated explicitly the limitation that we are not recapitulated our findings in human beta cells in the abstract and in the discussion section.

Abstract:

Despite obesity being a predisposing factor for pancreatic β -cell dysfunction and loss, the mechanisms underlying its negative effect on insulin-secreting cells remain poorly understood. In this study, we identified an islet-enriched long non-coding RNA (lncRNA), which we named β -cell function and apoptosis regulator (*β Faar*). *β Faar* was dramatically downregulated in the islets of the obese mice, and a low level of *β Faar* was necessary for the development of obesity-associated β -cell dysfunction and apoptosis in mice. Mechanistically, *β Faar* promoted the synthesis and secretion of insulin by upregulating islet-specific genes *Ins2*, *NeuroD1*, and *Creb1* through sponging *miR-138-5p*. In addition, using quantitative mass spectrometry, we identified TRAF3IP2 and SMURF1 as interacting proteins that are specifically associated with *β Faar*. We demonstrated that SMURF1 ubiquitin ligase activity is essential for TRAF3IP2 ubiquitination and activation of NF- κ B-mediated β -cell apoptosis. **Although the function of *β Faar* in human β cells is yet to be investigated**, our experiments provide direct evidence that dysregulated *β Faar* contributes to the development of obesity-induced β -cell injury and apoptosis.

Discussion:

.....Although the lncRNA *β FAAR (TRAF3IP2-AS1)* has been reported to be associated with tumors¹, and upregulated in cocaine abusers². Interestingly, the expression of human lncRNA *β FAAR (TRAF3IP2-AS1)* was predicted in human β cells by Akerman³. However, our study explore the role of *mus- β Faar* in β -cell function in mice, and the possibility that *has- β FAAR* regulates human β -cell function requires further research.....

2. *Further evidence corroborating an action on NF- κ B signalling would reinforce the conclusions of the manuscript, and could include measurement of the subcellular distribution of NF- κ B subunits, or phosphorylation of IK-B, after manipulation of *bFaar* expression.*

Response: Thank you for this important point. Following your suggestion, we have detected some major NF- κ B signaling targets (P65, IK-B, MnSOD, FAS, and iNOS) in the MIN6 cells transfected with *oe- β Faar* or *si- β Faar*. We found that silenced *β Faar* could active NF- κ B signaling. The similar trend was observed in the islets of *β Faar-KO* mice. We hope the explanations and changes above would strengthen our conclusions.

.....Of note, the induction of apoptosis of β -cells by overexpressed *Traf3ip2* was accompanied by

phosphorylation of p65 and IκB, and activation of some of the major targets of NF-κB signaling, such as MnSOD, Fas, and iNOS (Figure 7j). We also found that NF-κB signaling was stimulated in MIN6 cells transfected with *si-βFaar* and in the islets of *βFaar-KO* mice, and the opposite effect was observed with *βFaar* overexpression (Figure S7 c-e).....

Figure S7 The protein expression levels of P65, IκBα and some major NF-κB signaling targets (MnSOD, Fas, and iNOS) in the MIN6 cells transfected with *oe-βFaar* or *si-βFaar* (c), in the islets of *βFaar-KO* mice (d, *n* = 5) and in the islets of *len-βFaar* mice (e, *n* = 5).

3. What is the distribution of *βFaar* expression between beta and other islet cell types?

Response: Thank you for this important suggestion. As suggested, we have examined *βFaar* expression in the β cells, α cells and δ cells through Insulin/*βFaar*, Glucagon/*βFaar* or Somatostatin/*βFaar* double stain. The results showed that *βFaar* was almost enriched in the β cells. Immunofluorescence showed that *βFaar* expression was enriched in β cells (Figure S1e).....

Figure S1 (e) FISH analysis of *βFaar* expression level in the β cells, α cells and δ cells. The nuclei were stained with DAPI. Scale bar, 20 μm.

4. Can the authors shed more light on the intracellular distribution of *βFaar*?

Response: Thank you for this insightful comment. In our study, we used subcellular fractionation and RNA fluorescent *in situ* hybridization (RNA-FISH) assays to analyze the intracellular distribution of $\beta Faar$ in MIN6 cells and primary islets. We found that $\beta Faar$ was moderately more abundant in the cytoplasm than in the nucleus of MIN6 cells and primary islet cells (Figure 1g and f). Based on previous reports, subcellular fractionation and RNA-FISH analysis often used to determine the cellular localization of lncRNA^{4, 5, 6, 7}. Thus, currently we can't shed more methods to detect the intracellular distribution of $\beta Faar$ except subcellular fractionation and RNA-FISH assays. If reviewers can recommend other methods, we will be very happy to try.

Here we demonstrated that $\beta Faar$ promoted insulin transcription and secretion by upregulating islet-specific gene (*Ins2*, *NeuroD1* and *Creb1*) expression through sponging *miR-138-5p*. Moreover, reduced $\beta Faar$ inhibits to recruit E3 ubiquitin ligase SMURF1, then suppressing the ubiquitination and degradation of TRAF3IP2, in turn activating the NF- κ B signal pathway, and finally inducing β -cell apoptosis. Since *miR-138-5p*, SMURF1 and TRAF3IP2 (Figure 7n) were mainly located in cytoplasm, which was agreed with RNA fluorescent *in situ* hybridization (RNA-FISH) assays results that $\beta Faar$ was moderately more abundant in the cytoplasm than in the nucleus of MIN6 cells and primary islet cells. We have added some sentence to describe the intracellular distribution of $\beta Faar$.

Figure 1

Figure1 (g) $\beta Faar$ was enriched in the MIN6 cell and primary islet cell cytoplasm fraction. Levels of $\beta Faar$, *Gapdh* mRNA and *U6* small nuclear RNA in purified MIN6 cells and islet cells nuclear and cytoplasm fractions were detected by qRT-PCR. (h) FISH analysis of $\beta Faar$ in MIN6 cells and islet cells. The nuclei were stained with DAPI. Magnification: 40 x or 20 x, scale bar, 10 μ m or 20 μ m.

Results

.....Furthermore, ***β Faar* colocalized with TRAF3IP2 and SMURF1 in the cytoplasm by FISH/IF (Figure 7n)**. All these results confirmed the interaction of TRAF3IP2 and SMURF1 with *β Faar*. SMURF1, whose protein level was no significantly changed in the islets of HFD and *db/db* mice (Figure S7k, l), has the activity of ubiquitin E3 ligase in its HECT domain that mediates the ubiquitination of other proteins⁹.....

Discussion

..... Mechanistically, we first documented that *β Faar* is colocalized with TRAF3IP2 and SMURF1 in the cytoplasm and then showed the evidence that *β Faar* forms complexes with E3 ubiquitin ligases SMURF1 and interacts with its ubiquitination substrate TRAF3IP2. By facilitating the formation of these complexes, *β Faar* promotes the SMURF1-mediated ubiquitination of TRAF3IP2, increasing its degradation. These results uncover the role of a lncRNA, *β Faar*, as a platform for protein ubiquitination. Furthermore, our results adequately explain the mechanism by which the downregulation of *β Faar* induces β -cell apoptosis in obesity. The increase of TRAF3IP2 can activate NF- κ B pathway.....

Figure 7 (n) Colocalization of *β Faar* (red), TRAF3IP2 or SMURF1 (green) and INSULIN (pink) was visualized by FISH/IF assays. Scale bar, 20 μ m.

Minor

1. The method of overexpression or KO in vivo - intraductal infusion; <https://www.ncbi.nlm.nih.gov/pmc/articles/PMC4734891/> affects all pancreatic cells, including exocrine. What was the proportion of islets is affected overall, and how does this impact the interpretation of the data both in isolated islets and in vivo (glucose tolerance)? This may or may not be a problem, but surely it's better to have islet/beta cell-selective expression (insulin promoter/optimized viral serotype?)

Response: Thank you for this insightful comment and we apologize for the ambiguous formulation. In our study, A Rat insulin 2 promoter was replaced the ZsGreen1 of pHAGE-CMV-MCS-IZsGreen vector. Then *βFaar* or *Traf3ip2* overexpression plasmid was constructed into the pHAGE-CMV-Rat *Insulin2* promoter (pHAGE-CMV-RIP) vector using one step cloning kit. Moreover, we used Rat *insulin2* promoter to replace the EF-1-alpha core promoter of lentiCRISPRv2 puro vector using EcoR I and Xba I , then *βFaar-sgRNA* was constructed in lentiCRISPRv2-Rat *Insulin2* promoter (lentiCRISPRv2-RIP) vector. Rat Insulin 2 promoter sequence and sequence map of *len-βFaar*, *len-Traf3ip2* and *βFaar-sgRNA* were listed below. And we have added lentiviral production in the methods. And, we isolated islets and exocrine glands of control and *len-βFaar* mice, then tested the *βFaar* expression in the islets and exocrine glands. qRT-PCR result revealed that *βFaar* expression was only up-regulated in islets of *len-βFaar* mice (the result is shown below). Moreover, we found glucagon level showed no significantly change in serum of the *len-βFaar* mice and *βFaar-KO* mice (Figure S31). Taken together, these data showed that lentivirus *βFaar* injected into mice through the pancreatic intra-ductal infusion was mainly regulated β cells function. We hope the explanations and changes above would make you and other readers much easier to understand our manuscript.

Methods

Lentiviral production

βFaar or *Traf3ip2* overexpression plasmid was constructed in the pHAGE-CMV-Rat *Insulin2* promoter (pHAGE-CMV-RIP) vector using the One Step Cloning Kit (C113-02, Vazyme Biotech Ltd., Nanjing, China). *βFaar-sgRNA* was constructed in lentiCRISPRv2-Rat *Insulin 2* promoter (lentiCRISPRv2-RIP) vector using *BsmB I*. Lentiviral particles were produced by transient transfection of HEK293T cells with the packaging plasmids pMD2.G (Addgene #12259) and psPAX2 (Addgene #12260), together with the lentiviral *βFaar*- or *Traf3ip2*- overexpression vector (*len-βFaar*, *len-Traf3ip2*) using Lipofectamine 3000 (Invitrogen, Carlsbad, CA, USA). Lentiviral particles carrying *βFaar-sgRNA*-expressing vectors (*βFaar-KO*) were also obtained from HEK293T cells using the same method. The culture medium was collected 48 h after transfection of HEK293T, and lentiviral particles were concentrated by ultracentrifugation. For each construct, the lentiviral titer was determined by infection of HEK293T cells and FACS analysis.

Results

.....The ectopic expression of *βFaar* did not change the islet structure and the content of glucagon (Figure S3k, l).....

Figure S3

Figure 3 (k) Pancreatic sections were stained with insulin and glucagon antibody using immunofluorescence. Magnification: 20 x, Scale bars: 20 μ m ($n = 3$). (l) Glucagon content was tested by ELISA in the primary islet ($n = 9$).

Figure legend: The expression levels of $\beta Faar$ in the islets and exocrine glands of control and *len- $\beta Faar$* mice ($n = 3$). *** $p < 0.001$.

Rat Insulin 2 promoter sequence:

GGATCCCCCAACCACTCCAAGTGGAGGCTGAGAAAGGTTTTGTAGCTGGGTAGAGTA
 TGTACTAAGAGATGGAGACAGCTGGCTCTGAGCTCTGAAGCAAGCACCTCTTATGGA
 GAGTTGCTGACCTTCAGGTGCAAATCTAAGATACTACAGGAGAATACACCATGGGGC
 TTCAGCCCAGTTGACTCCCGAGTGGGCTATGGGTTTTGTGGAAGGAGAGATAGAAGAG

AAGGGACCTTTCTTCTTGAATTCTGCTTTCCTTCTACCTCTGAGGGTGAGCTGGGGTCT
 CAGCTGAGATGAGGACACAGCTATCAGTGGGAAGTGTGAAACAACAGTTCAAGGGA
 CAAAGTTACTAGGTCCCCAACAACTGCAGCCTCCTGGGGAATGATGTGGAAAATGC
 TCAGCCAAGGACAAAGAAGGCCTCACCTCTCTGAGACAATGTCCCCTGCTGTGAAC
 TGGTTCATCAGGCCACCCAGGAGCCCCTATTAAGACTCTAATTACCCTAAGGCTAAGT
 AGAGGTGTTGTTGTCCAATGAGCACTTCTGACAGACCTAGCACCAGGCAAGTGTGG
 AAAGTGCAGCTTCAGCCCCTCTGGCCATCTGCTGATCCACCCTTAATGGGACAAACAG
 CAAAGTCCAGGGGTCAGGGGGGGGTGCTTTGGACTATAAAGCTAGTGGGGATTCACT
 AACCCAGCCCTAA (703 bp).

Sequence map of *len-βFaar*, *len-Traf3ip2* and two *βFaar*-sgRNAs:

2. Are other DNMT3a, -3b targets affected by palmitate?

Response: Thank you for this critical suggestion. In our study, we did find an interesting phenomenon that DNMT3a and DNMT3b are increased by palmitate. Multiple studies have

highlighted changes in DNA methylation at loci associated with β cell function from subjects with type 2 diabetes^{8, 9, 10}. Due to the changes of DNMT3a and DNMT3b induced by palmitate, genes with CpG-rich regions in promoter may be affected. For example, in our previous study, we found that obesity induced NKX6.1 methylation was due to up regulation of Dnmt3a¹¹. Here, we found that Nkx6.1 as Dnmt3a targets affected by palmitate too (The data is shown below).

Figure legend: MIN6 cells were incubated with 0.5 mmol/l palmitate for 48 h, qPT-PCR and Western blot were performed to test NKX6.1 expression. *** $p < 0.001$.

3. *There are issues throughout with English and grammar that require attention.*

Response: Thank you for this critical suggestion. We have asked native English speakers to rewrite the manuscript that the reviewers thought we lacked details or clarity. Editorial certificate was listed below.

Reviewer #2 (Remarks to the Author):

RE: NCOMMS-20-35111-T

*The paper described the discovery of \square Faar, a lncRNA, that is highly expressed in the mouse pancreatic insulin-secreting \square cells. Using both in vitro and in vivo approaches the authors demonstrated that \square Faar positively regulates insulin biogenesis and protects \square cells from obesity-induced apoptosis. Mechanistically, the authors provided compelling evidence that \square Faar stimulates insulin biosynthesis by sponging miR-138-5p whereby derepressing key transcription factors *Ins2*, *Creb1* and *NeuroD1* known to regulate insulin transcription. On the other hand, the authors showed that \square Faar acted as an assembly scaffold for TRAF3IP2 and SMURF1 to induce ubiquitination and degradation of TRAF3IP2. Specifically, TRAF3IP2 activated NF- κ B signaling to induce \square cell apoptosis and \square Faar protected \square cells by downregulating TRAF3IP2. The paper is nicely written, the mode of action of \square Faar is novel and intriguing, and the conclusions are overall well supported by the data.*

However, I have a major concern about the conservation of \square Faar between human and mouse. A common rule of thumb for two sequences considered to be homologous is that they are more than 30% identical over their entire lengths. As shown in Supplementary Fig. 1c-f, the overall identity between the mouse \square Faar and the putative human \square Faar is less than 30%, indicating that they are not conserved based on the sequence alone. A careful definition of conservation would require more advanced methods involving statistic models such as PhyloP and PhyloHMM. Numerous lncRNAs identified so far in the mouse have been shown to have no functional counterparts in the human. It is very important that the authors provide evidence on which of the putative human \square Faar RNA(s) are actually expressed in human pancreatic cells and if so, whether they perform functions similar to the mouse \square Faar. Human pancreatic cell lines should be available (Scharfmann et al, The supply of chain of human pancreatic cell lines. JCI, 2019, 129(9):3511-3520). Otherwise, the statement “characterize the \square Faar as a potential target for treatment of obesity-associated diabetes” (lines 35-36) is invalid, and the translational significance of this work is severely diminished.

Response: Thank you for your kind advice. In our previous study, the sequence of *Mus- β Faar* (only one transcript variant) and *Has- β FAAR* (four transcript variants) were obtained from NCBI, then we used MEGA 6 and DNAMAN to analyze the conservation between *Mus- β Faar* and *Has- β FAAR*. As

showed in Figure S1 c-f, the overall identity between the mouse *Faar* and the human *βFAAR* was 18.45%, 17.6%, 35.14% and 20.86%. Since the sequence length of *Mus-βFaar* and *Has-βFAAR* is large different, our method is not really reasonable. According to your critical suggestion and previous reports^{12,13}, we used PhyloP, which is a more advanced methods involving statistic models, to analyze the overall identity between the mouse *βFaar* and the human *βFAAR* (Figure S1b, listed below). The result revealed that the sequence between *mus-βFaar* and *has-βFAAR* showed some conservation.

Secondly, it is better to study *βFaar* on the β cells function using mouse β cells and human β cells. However, it is hard to obtain human pancreatic cell lines, we mainly investigate *mus-βFaar* on the β cells function. As suggested, we email to Scharfmann for requiring EndoC- β H1 cells (Scharfmann et al, The supply of chain of human pancreatic cell lines. JCI, 2019, 129(9):3511-3520). However, due to COVID-19, EndoC- β H1 cells can't be successfully shipped in China, and it is hard to obtain pancreatic islets. These factors restrict us to investigate whether human *βFAAR* perform functions similar to the mouse β *Faar* in human β cells.

Since we proved that *Mus-βFaar* expression level was significantly decreased in the islets and WAT. In our further study, we collected some white adipose tissue (WAT), including 70 obesity patients (BMI > 25) and 15 normal individuals (20 < BMI < 25) from the Zhongda Hospital, Affiliated to southeast University (Nanjing, China). Firstly, we found that *has-βFAAR* transcript variant 3 in the WAT of normal individuals showed a higher expression level (conservation is 35.14% analyzed by MEGA 6 and DNAMAN). Moreover, we found that expression levels of four human *βFAAR* transcript variants were markedly decreased in the WAT of obesity patients (BMI > 25), among them, the expression levels of *βFAAR* of V3 and V4 displayed lower. And based on our previous study, the overall identity between the mouse *βFaar* and the V3 and V4 of human *βFAAR* were 35.14% and 20.86% respectively analyzed by MEGA 6 and DNAMAN. Though we have not investigated *has-βFAAR* function in human β cells, we did prove that obesity induced *has-βFAAR* down-regulated in the WAT. Of note, insulin resistance and β -cell failure contribute to the development of T2DM. Adipose tissue is a critical target tissue of insulin. Taken together, we speculate that *has-βFAAR* may play a significantly role in diabetes. We will systematically research the effect of *has-βFAAR* on the β cells function once we obtain human pancreatic cell lines.

According to your suggestion, we have deleted our writing “and characterize the *βFaar* as a potential target for the treatment of obesity-associated diabetes.” (lines 35-36).

..... *Mus-βFaar* has only one variant and *Has-βFAAR* has four variants. The UCSC genome browser view of the *Mus-βFaar* shows location conserved with human *βFAAR* by PhyloP statistical model (Figure S1b).....

Figure S1 (b) UCSC Genome Browser scheme of the human *βFAAR* variants 1–4 (red) and mouse *βFaar*. And the conservation between mouse *βFaar* and human *βFAAR* (Hg38 genome build).

Table 1 Clinical characteristics of the patients with obese patients and normal individuals

	Obesity	Normal	Total
Number (male/female)	70 (35/35)	15 (7/8)	85 (42/43)
Age (years)	32.84±9.21	31.00±10.19	32.48±9.38
BMI	37.52±5.97	21.84±0.64	36.28±7.34

Figure legend:

(a) Schematic diagram of primer design for detecting four variants of human β FAAR. (b) qPCR was performed to test the β FAAR (V1-V4) expression level in the white adipose tissue (WAT) of normal individuals. (c-f) qPCR was performed to detect V1 (c), V2 (d), V3 (e) and V4 (f) expression in the

WAT of obesity patients compared to normal individual (n = 15/70). All experiments above were performed in triplicates, and each group contained three batches of individual samples. The *p*-values by two-tailed unpaired Student's *t* test. Data represent the mean ± SEM. ****p* < 0.005.

Minor points:

1. The evidence for DNMT3a and DNMT3b involved in obesity-induced $\square\square$ Faar hypermethylation and downregulation is weak. Increased DNMT3a and DNMT3b expression was only shown in the pancreatic islets from *db/db* mice. The authors need to show whether this is also the case for other obese models, such as HFD. Otherwise, the authors should remove DNMT3b and DNMT3a from Fig. 8. Without data from human \square cells the authors should replace the human cartoon with a mouse.

Response: Thank you for this insightful comment. We have detected DNMT3a and DNMT3b in the HFD mice. The result revealed that DNMT3a and DNMT3b were increased, while DNMT1 was not changed significantly. Moreover, according to your good suggestion, we have replaced the human cartoon with a mouse. We hope the explanations and changes above would make you and other readers much easier to understand our manuscript.

..... We found that the protein levels of DNMT3a and DNMT3b were significantly higher in the islets of diabetic *db/db* mice and HFD-fed mice than in controls, but the protein level of DNMT1 was not significantly changed (Figure 2e, f).....

Figure 2 (e-f) The protein levels of DNMT1, DNMT3a and DNMT3b in the islet of *db/db* mice (e) and HFD mice (f).

Figure 8 Schematic illustration for the mechanism of obesity-induced reduction of $\beta Faar$ impairing insulin biosynthesis and exacerbated β -cells apoptosis. During obesity, $\beta Faar$ is downregulated by DNA hypermethylation, suppressing insulin biosynthesis and secretion by downregulating islet-specific gene (*Ins2*, *NeuroD1* and *Creb1*) expression through sponging *miR-138-5p*. Moreover, reduced $\beta Faar$ inhibits to recruit E3 ubiquitin ligase SMURF1, then suppressing the ubiquitination and degradation of TRAF3IP2, activating the NF- κ B signal pathway, and finally inducing β -cell apoptosis.

2. In Fig. S3a, and b, the 450-fold increase in the level of $\square Faar$ by *oe- $\square Faar$* was an exaggeration. I suspect it was likely an artifact from a contamination of the transfected plasmid DNA (i.e., incomplete DNase digestion prior to RT-qPCR). Cells transfected with the $\square Faar$ plasmid would retain the plasmid and that a small fraction of it would escape DNase digestion during RNA purification and be amplified by RT and the highly sensitive PCR. The authors can test this possibility by putting through RNA purification 1/5-1/10 amounts of the $\square Faar$ plasmid DNA used to transfect the cells, followed by RT-qPCR. I am sure they will see strong $\square Faar$ signals that at least in part actually come from the contaminated plasmid DNA.

Response: Thank you for this important advice. As suggested, we used DNase digestion during

RNA purification, after amplified by RT, qPCR was used to test $\beta Faar$ expression. The results showed that $\beta Faar$ expression level was increased about 160-fold in the primary islets and about 200-fold in the MIN6 cells.

Figure S3

Figure S3 $\beta Faar$ overexpression plasmid (oe- $\beta Faar$) and $\beta Faar$ smart silence (si- $\beta Faar$) were transfected into primary islet cells and MIN6 cells for 48 h. Then the transfection efficiencies of $\beta Faar$ were analyzed by qRT-PCR in primary islets (a) and in MIN6 cells (b).

3. Lines 140-142: the sentence “We next confirmed that $\beta Faar$ was re-expressed in MIN6 cells and primary islets after DNA demethylation treatment with 5-Aza-dC in a dose-dependent manner (Figure 2c)” is a bit misleading. It gave one the impression that $\beta Faar$ expression was first inhibited (by palmate or cytokines) and then derepressed by 5-Aza-dC. It should be rephrased to something like “Treatment of MIN6 cells and primary islets with 5-Aza-dC increased $\beta Faar$ expression in a dose-dependent manner”.

Response: Thank you for your suggestion, and we apologize for the ambiguous formulation. We have corrected it.

.....Next, we found that the treatment of MIN6 cells and primary islets with 5-Aza-dC increased $\beta Faar$ expression in a dose-dependent manner (Figure 2c).....

4. Line 175: “Camp” should be “cAMP”.

Response: Thank you for your careful reviewing and we are sorry for our incorrect writing. We have corrected it.

Reviewer #3 (Remarks to the Author):

In this manuscript, the authors identified a lncRNA (named β Faar) dramatically downregulated in the islets of obese mice and necessary for β -cell dysfunction and apoptosis. As to the mechanisms referring to β -cell apoptosis during obesity, the authors showed that β Faar could negatively regulate expression of TRAF3IP2, a NF- κ B pathway activator, and provided a scaffold to boost the interaction between TRAF3IP2 and Smurf1, a known HECT type E3 ligase. This conclusion was highlighted by the authors in the abstract, emphasizing a potential important role of Smurf1 involved in NF- κ B regulation and β -cell dysfunction. It is a new knowledge about Smurf1 function, which is impressive for Smurf1 biology. However, these data mainly derived from cells with genes artificial overexpression or knockdown manipulation, it's preliminary and not convincing. More in-vivo evidence of β Faar-TRAF3IP2-Smurf1 regulation and related β -cell function should be provided.

Response: We appreciate the reviewer for her/his encouraging comment. We followed your suggestion and provided more convinced evidences to strengthen our conclusions. For example, we have tested SMURF1 expression level in the islets of HFD mice and *db/db* mice. Moreover, we have used Co-IP to clarify β Faar-TRAF3IP2-Smurf1 regulation *in vivo*. We hope these experiments would strengthen our conclusions.

1. *First, the authors need to manipulate β Faar or TRAF3IP2 or Smurf1 in vivo, by the way of overexpressing or knocking down, to recapitulate the conclusion achieved in cells. For example, β Faar should be decreased in mice to see if TRAFIP2 was increased, to examine if NF- κ B was activated, to test if β -cell apoptosis was observed. Further, β -cell area, or insulin secretion activity should be examined.*

Response: Thank you for your critical reviewing. Firstly, according to your suggestions, we have tested the protein levels of TRAF3IP2 and SMURF1 in the islets of *len- β Faar* mice and *β Faar-KO* mice. We found that TRAF3IP2 was decreased in the islets of *len- β Faar* mice and increased in *β Faar-KO* mice (Figure 7b, c), while β Faar has no effect on SMURF1 (Figure S7p, q). Interestingly, knockdown of β Faar induces increasing of TRAF3IP2 accompanied by phosphorylation of p65 and I κ B, and activation of some major NF- κ B signaling targets (MnSOD, Fas, and iNOS) (Figure S7c, d, e). It suggests that decreased β Faar increases *Traf3ip2* expression to activate NF- κ B signaling.

Secondly, we apologize for the ambiguous formulation. In our study, we have tested the β -cell apoptosis (Figure 4g), β -cell mass (Figure S4c) and insulin secretion (Figure 3 l, m) in *len- β Faar* mice and *β Faar-KO* mice. Moreover, we have found that *len-Traf3ip2* mice could increase positive TUNEL β cell while combined injection of *len- β Faar* and *len-Traf3ip2* restored β -cell apoptosis compared to *len-Traf3ip2* mice (Figure 7i). Taken together, these results revealed that overexpression *β Faar* could decrease TRAF3IP2 protein level *in vivo* and *in vitro*, then suppressing NF-kB pathway, and finally improving β cell function.

To facilitate your check, the main correction in revision was listed below.

..... We found that the protein level of TRAF3IP2 was increased in *β Faar* knockdown MIN6 cells and decreased in *β Faar* overexpressing MIN6 cells (Figure 7a), but had no effect on its mRNA (Figure S7b). Additionally, TRAF3IP2 protein level was increased in the islets of *β Faar-KO* mice and decreased in the islets of *len- β Faar* mice (Figure 7b, c).....

Figure 7 Western blot was used to test the TRAF3IP2 protein levels in the islets of *len- β Faar* mice (b) and *β Faar-KO* mice (c, $n = 5$).

.....Finally, the overexpression or knockdown *β Faar* did not modulate SMURF1 expression (Figure S7o-q), and the overexpression of *Smurf1* decreased TRAF3IP2 protein level (Figure S7r).....

Figure S7

Figure S7 The SMURF1 protein levels in the MIN6 cells transfected with *si-βFaar* or *oe-βFaar* (o), in the islets of HFD mice (p) and *db/db* mice (q, n = 5).

..... In addition, insulin synthesis and insulin sensitivity to glucose were increased in the islets derived from *len-βFaar* mice (n = 5) (Figure 3l, m), while an opposite result was obtained in *βFaar-KO* mice.....

Figure 3 (l) Relative *Ins1* and *Ins2* expression levels *in vivo* (n = 3). (m) Static insulin secretion of islets (n = 5) at indicated glucose concentrations.

..... To verify whether downregulation of *βFaar* also causes β -cell apoptosis *in vivo*, we mimicked the obesity-associated decrease in *βFaar* expression by knocking down *βFaar* in normal C57BL/6J mice through the pancreatic ductal infusion of *βFaar-KO*. The downregulation of *βFaar* significantly increased the level of cleaved caspase-3 and BAX while decreasing the level of procaspase-3 and BCL-2 expression (Figure 4e). Opposite effects were achieved with the overexpression of *βFaar* (Figure 4f). Furthermore, the inhibition of *βFaar* in normal mice led to an increase in the number of TUNEL-positive β -cells (Figure 4g). Next, we assessed the contribution of increased *βFaar* expression to the protection of islets by overexpressing *βFaar* in normal mice. β -cell mass in mice treated with *len-βFaar* was slightly higher than in control mice (Figure S4c). These findings indicate that the knockdown of *βFaar* alone is sufficient to impair the function of β -cell and induce their apoptosis.....

Figure 4 (e-f) The protein levels of Pro-Caspase 3, Cleaved Caspase 3, BCL-2 and BAX in the islet of $\beta Faar$ -KO mice (e, $n = 5$) and len - $\beta Faar$ mice (f, $n = 5$). (g) The TUNEL positive β cells in the islets of $\beta Faar$ -KO mice and len - $\beta Faar$ mice. Magnification: 20 x, scale bar, 20 μ m.

Figure S4 (c) The β -cell mass of $\beta Faar$ -KO mice and len - $\beta Faar$ mice was measured.

2. Also, the authors showed evidence in cells that $\beta Faar$ forms complexes with *Smurf1* and *TRAF3IP2*, and promotes the interaction and ubiquitination of *TRAF3IP2* by *Smurf1*. I strongly suggest the authors to examine the interplay (interaction and ubiquitination assays) between *TRAF3IP2* and *Smurf1* *in vivo* by overexpressing $\beta Faar$ in mice model.

Response: Thank you for your meaningful comments. According to your advice, we have examined the interplay (interaction and ubiquitination assays) between *TRAF3IP2* and *Smurf1* *in vivo* by overexpressing $\beta Faar$ in mice model. We isolated islets of len - $\beta Faar$ mice and control mice. We found that overexpression of $\beta Faar$ could enhance the interaction of *TRAF3IP2* and *SMURF1* in

in vivo (Figure 7p), and facilitated TRAF3IP2 ubiquitination (Figure 7u). Moreover, we isolated primary islets from *len-shSmurf1* mice and both treated *len-βFaar* and *len-shSmurf1* mice, Co-IP result revealed that TRAF3IP2 ubiquitination was inhibited after silencing SMURF1, even in the presence of elevated *βFaar* levels (Figure 7w). We hope the explanations and changes above would strengthen our conclusions.

To facilitate your check, the main correction in revision was listed below.

.....Interestingly, the interaction of TRAF3IP2 and SMURF1, assessed by Co-IP followed by WB analysis, was facilitated by the overexpression of *βFaar in vitro* and *in vivo* (Figure 7o, p).....

.....Next, we determined whether *βFaar* affects the ubiquitination of associated TRAF3IP2. Immunoprecipitation of TRAF3IP2 followed by ubiquitin detection by Western blotting revealed that the overexpression of *βFaar* increased the pool of ubiquitinated TRAF3IP2 *in vitro* and *in vivo*, identified as high-molecular-weight discrete or diffuse bands (Figure 7t, u). Additionally, TRAF3IP2 ubiquitination was inhibited after silencing SMURF1, even in the presence of elevated *βFaar* levels (Figure 7v, w). Taken together, these data demonstrate that *βFaar* serves as an assembly scaffold that facilitates the interaction of E3 ligases (SMURF1) with the ubiquitination target TRAF3IP2.....

Figure 7

Figure 7 (p) *βFaar* promoted SMURF1 binding to TRAF3IP2. Cell lysates from primary islets from *len-βFaar* mice ($n = 5$), treated with 20 $\mu\text{mol/l}$ MG132 for 6 h, were subjected to co-immunoprecipitation with the antibody against SMURF1 (IP: SMURF1) or TRAF3IP2 (IP: TRAF3IP2) followed by Western blotting. (u) *βFaar* promotes TRAF3IP2 ubiquitination. primary islets isolated from *len-βFaar* mice ($n = 5$), after treatment with 20 $\mu\text{mol/l}$ MG132 for 6 h, the cells were subjected to co-immunoprecipitation with the antibody against TRAF3IP2 (IP: TRAF3IP2) followed by western blotting. (w) SMURF1 promotes TRAF3IP2 ubiquitination. Islets isolated from *len-shSmurf1* and *len-βFaar* and *len-shSmurf1* mice ($n = 5$), then cells were treated with 20 $\mu\text{mol/l}$

MG132 for 6 h. The cells were subjected to co-immunoprecipitation with the antibody against TRAF3IP2 (IP: TRAF3IP2) followed by western blotting.

3. In Fig.7, the authors gave a detailed information about the interplay between β Faar and TRAF3IP2. However, when *Smurf1* is introduced into the three-part interaction, how *Smurf1* is regulated by β Faar, how *Smurf1* expression is changed under HFD treatment or in *db/db* mice, and how *Smurf1* is involved in NF- κ B pathway are missing. The authors should design a series of experimental data in detail to tell us whether *Smurf1* is truly involved in this regulation.

Response: We appreciate the reviewer for this insightful comment. Here, we found that SMURF1 expression was not changed under HFD mice and *db/db* mice (Figure S7k, l, $n = 5$). And we identified that overexpression or knockdown of β Faar could not alter SMURF1 expression (Figure S7o), the similar result was also observed in the islets of *len- β Faar* and *β Faar-KO* mice (Figure S7p and q). Since obesity and overexpression β Faar can't change SMURF1 expression. These data revealed that *Smurf1* is not directly involved in NF- κ B pathway in our study.

While SMURF1 is predicted with higher confidence as a primary E3 ligase for TRAF3IP2 in UbiBrowser database (<http://ubibrowser.ncpsb.org/ubibrowser>) (Figure S7i), and we have determined that β Faar could enhance the interaction of TRAF3IP2 and SMURF1 both *in vitro* and *in vivo* (Figure 7o, p), and β Faar serves as an assembly scaffold that facilitates the interaction of E3 ligases (SMURF1) with ubiquitination targets TRAF3IP2, finally inhibited NF- κ B pathway. We hope the explanations and changes above would make you and other readers much easier to understand our manuscript.

To facilitate your check, the main correction in revision was listed below.

.....All these results confirmed the interaction of TRAF3IP2 and SMURF1 with β Faar. SMURF1, whose protein level was no significantly changed in the islets of HFD and *db/db* mice (Figure S7k, l), has the activity of ubiquitin E3 ligase in its HECT domain that mediates the ubiquitination of other proteins¹⁴. Moreover, the endogenous Co-IP assay showed that SMURF1 was an associated protein of TRAF3IP2 (Figure S7m, n). Interestingly, the interaction of TRAF3IP2 and SMURF1, assessed by Co-IP followed by WB analysis, was facilitated by the overexpression of β Faar *in vitro* and *in vivo* (Figure 7o, p). Finally, the overexpression or knockdown β Faar did not modulate SMURF1 expression (Figure S7o-q), and the overexpression of *Smurf1* decreased TRAF3IP2 protein level (Figure S7r). The identification of these interactions allowed us to

hypothesize that $\beta Faar$ functions in protein ubiquitination and protein degradation.

Figure S7 The protein levels of SMURF1 in the islets of HFD mice (k, $n = 5$) and db/db mice (l, $n = 5$). The protein levels of SMURF1 in the MIN6 cells transfected with $si-\beta Faar$ or $oe-\beta Faar$ (o), in the islets of $len-\beta Faar$ mice (p, $n = 5$) and $\beta Faar$ -KO mice (q, $n = 5$). All experiments above were performed in triplicates, and each group contained three batches of individual samples.

- Zan, X. Y. & Li, L. Construction of lncRNA-mediated ceRNA network to reveal clinically relevant lncRNA biomarkers in glioblastomas. *Oncology letters*. **17**, 4369-4374 (2019).
- Bannon, M. J. *et al.* Identification of long noncoding RNAs dysregulated in the midbrain of human cocaine abusers. *J Neurochem*. **135**, 50-59 (2015).
- Akerman, I. *et al.* Human Pancreatic beta Cell lncRNAs Control Cell-Specific Regulatory Networks. *Cell Metab*. **25**, 400-411 (2017).
- Ou, C. *et al.* Targeting YAP1/LINC00152/FSCN1 Signaling Axis Prevents the Progression of Colorectal Cancer. *Advanced science (Weinheim, Baden-Wurtemberg, Germany)*. **7**, 1901380 (2020).
- Hu, Y. W. *et al.* Long noncoding RNA NEXN-AS1 mitigates atherosclerosis by regulating the actin-binding protein NEXN. *J Clin Invest*. **129**, 1115-1128 (2019).
- Singer, R. A. *et al.* The Long Noncoding RNA Paupar Modulates PAX6 Regulatory Activities to Promote Alpha Cell Development and Function. *Cell Metab*. **30**, 1091-1106 e1098 (2019).
- Chen, F. *et al.* Extracellular vesicle-packaged HIF-1 α -stabilizing lncRNA from tumour-associated macrophages regulates aerobic glycolysis of breast cancer cells. *Nat Cell Biol*. **21**, 498-510 (2019).
- Dayeh, T. *et al.* Genome-wide DNA methylation analysis of human pancreatic islets from type 2 diabetic and non-diabetic donors identifies candidate genes that influence insulin secretion. *PLoS Genet*. **10**, e1004160 (2014).
- Yang, B. T. *et al.* Insulin promoter DNA methylation correlates negatively with insulin gene expression and positively with HbA(1c) levels in human pancreatic islets. *Diabetologia*. **54**, 360-367

(2011).

10. Yang, B. T. *et al.* Increased DNA methylation and decreased expression of PDX-1 in pancreatic islets from patients with type 2 diabetes. *Mol Endocrinol.* **26**, 1203-1212 (2012).
11. Zhang, F. F. *et al.* Obesity-induced reduced expression of the lncRNA ROIT impairs insulin transcription by downregulation of Nkx6.1 methylation. *Diabetologia.* **63**, 811-824 (2020).
12. Sallam, T. *et al.* Transcriptional regulation of macrophage cholesterol efflux and atherogenesis by a long noncoding RNA. *Nat Med.* **24**, 304-312 (2018).
13. Ramani, R., Krumholz, K., Huang, Y. F. & Siepel, A. PhastWeb: a web interface for evolutionary conservation scoring of multiple sequence alignments using phastCons and phyloP. *Bioinformatics.* **35**, 2320-2322 (2019).
14. Zhu, H., Kavsak, P., Abdollah, S., Wrana, J. L. & Thomsen, G. H. A SMAD ubiquitin ligase targets the BMP pathway and affects embryonic pattern formation. *Nature.* **400**, 687-693 (1999).

REVIEWER COMMENTS

Reviewer #1 (Remarks to the Author):

The reviewers have dealt with my concerns appropriately, and the manuscript has been strengthened as a result. I have no further comments.

Reviewer #2 (Remarks to the Author):

The authors have successfully addressed some of my concerns with additional experiments and text editing. However, one major concern remains: the relevance of their findings to human islets/beta-cells. In their revised manuscript the authors provided new data showing that the expression of two human BFaar variants decreases in white adipose tissue of obese individuals as compared to controls. Such an alternative approach does not really address the relevance issue, because all they have shown is a simple correlation and in another type of tissue (WAT vs pancreas). Evidence that overexpression/knockdown of a human BFaar variant in human beta cells leads to altered cell function will significantly strengthen this paper.

Reviewer #3 (Remarks to the Author):

My major concerns about this study, especially the part of β Faar-Smurf1-TRAF3IP2 axis in the regulation of β cell apoptosis have been resolved.

Dear reviewers:

Thank you very much for your comments and advice to our manuscript entitled “**Obesity-inhibited LncRNA *βFaar* regulates islet β -cell function and survival**”. We completely accept your recommendation and fully agree that these recommendations. We have revised the manuscript very carefully according to the suggestion. To clearly present the response, the comments are shown in *italics* and our responses are shown in **blue font**. A thorough, point-by-point response to each point was raised, and a word file of the revised manuscript with all changes labelled in **red font** has been uploaded. If you have any further questions about the revision, please do not hesitate to contact us.

Best regards,

Liang Jin

Comments:

Reviewer #2 (Remarks to the Author):

*The authors have successfully addressed some of my concerns with additional experiments and text editing. However, one major concern remains: the relevance of their findings to human islets/beta-cells. In their revised manuscript the authors provided new data showing that the expression of two human *BFAAR* variants decreases in white adipose tissue of obese individuals as compared to controls. Such an alternative approach does not really address the relevance issue, because all they have shown is a simple correlation and in another type of tissue (WAT vs pancreas). Evidence that overexpression/knockdown of a human *BFAAR* variant in human beta cells leads to altered cell function will significantly strengthen this paper.*

Response: Thanks for your positive comments. As suggested, we have purchased EndoC- β H5 cells from Univercell Biosolutions (France, Invoice was listed below). However, due to COVID-19, EndoC- β H5 cells can't be successfully arrived in China within three months. Fortunately, we have obtained human primary islets from Tianjin First Central Hospital. All procedures using human islets were approved by the Research Ethics Committee of the Tianjin First Central Hospital¹. And we have overexpressed or knockdown of *β FAAR* variant in human islets to research the function of *β FAAR*.

Firstly, we found that *has- β FAAR* transcript variant 4 in the human islets showed a higher expression level (Figure 1b and Figure S1c), thus we choose variant 4 to research *β FAAR* function. Then human islets were exposed to pathophysiological concentrations of palmitate, and pro-inflammatory cytokines. And result showed that the expression of *β FAAR* decreased in the presence of palmitate (0.5 mmol/l) (Figure 2c) and pro-inflammatory cytokines (Figure 2d), but not upon incubation with high glucose (33.3 mmol/l) (Figure S2b). This mode of action is consistent with the results in mice.

Next, to explore the potential role of *β FAAR* in regulating β -cell function, *β FAAR* was overexpressed or knockdown in human islets (Figure S3g), we have demonstrated that overexpression of *β FAAR* in human islets could increase INSULIN gene expression (Figure 3d), insulin content (Figure 3e) and insulin secretion (Figure 3f). Moreover, our results showed overexpression of *β FAAR* decreased the number of apoptotic cells by TUNEL assays (Figure 4c). These results are consistent with the phenomenon observed in mice.

In summary, these data indicate that *has- β FAAR* also can improve β -cell function, these

phenomena are the same as *mus-βFaar*. We hope the explanations and changes above would strengthen our conclusions.

Figure legend: The invoice of EndoC-βH5 cells obtained from Univercell Biosolutions.

To facilitate your check, the main correction in revision was listed below.

.....And we found that *has-βFAAR* transcript variant 4 in the human islets showed a higher expression level (Figure 1b and Figure S1c).....

Figure 1b

Figure 1 (b) qRT-PCR was performed to test the β FAAR (V1-V4) expression in the human islets.

Figure S1c

Figure S1 (c) Schematic diagram of primer design for detecting four variants of human β FAAR.

.....To determine the possible reason for the changes in β Faar expression in the islets of obese mice, we exposed MIN6 cells and normal mouse islets to glucose, pathophysiological concentrations of palmitate, and pro-inflammatory cytokines. The expression of β Faar decreased in the presence of palmitate (0.5 mmol/l) (Figure 2a) and pro-inflammatory cytokines (Figure 2b), but not upon incubation with high glucose (33.3 mmol/l) (Figure S2a). The same results were also observed in the huma islets (Figure 2c, d, and Figure S2b).....

Figure 2 The expression levels of β FAAR in human islets incubated with 0.5 mmol/l palmitate (c) and a combination of interleukin-1 β (IL-1 β , 5 ng/ml) and tumor necrosis factor- α (TNF- α , 30 ng/ml, d).

Figure S2 (b) Human islets were incubated with 2.5 mmol/l or 33.3 mmol/l glucose for 48 h and qRT-PCR was performed to examine the β FAAR levels.

.....Then β FAAR was overexpressed or knockdown in human islets(Figure S3g), we also found that overexpression of β FAAR in human islets could increase INSULIN gene expression (Figure 3d), insulin content (Figure 3e) and insulin secretion (Figure 3f).....

Figure 3

Figure 3 INSULIN gene expression (d), insulin content (e) and insulin secretion (f) in the human islets transfected with *oe-βFAAR* or *sg-βFAAR*.

Figure S3g

Figure S3 (g) The expression levels of *βFAAR* in the human islets transfected with *oe-βFAAR* or *sg-βFAAR*.

.....To identify the effect of *βFaar* on maintaining β-cell mass, we modified its expression in MIN6 cells. The CCK-8 assay showed that silencing *βFaar* reduced cell viability (Figure S4a). Moreover, the downregulation of *βFaar* in MIN6 cells, mimicking the conditions encountered in obesity, increased the number of apoptotic cells (Figure 4a). Similar results were obtained by the TUNEL assay in primary cultures of mice islets, in human islets and in MIN6 cells (Figure 4b, c and Fig.S4b).....

Figure 4c

Figure 4 (b-c) The both INSULIN and TUNEL positive cells were detected by immunofluorescence in the mice islets (b) and in the human islets (c). Magnification: 40 x or 20 x, scale bar, 10 μ m or 20 μ m.

Methods

Isolation and culture of primary islet cells

Human islets were provided from Tianjin First Central Hospital. All procedures using human islets were approved by the Research Ethics Committee of the Tianjin First Central Hospital¹. High purity islets (>80%) were collected and cultured in CMRL-1066 medium (Corning, Manassas, VA, USA), supplemented with 10% Human Serum Albumin (Baxter, Vienna, Austria), 100 U/mL penicillin and 100 μ g/mL streptomycin at 37°C in 5% CO₂.

Insulin secretion assay

Human pancreatic islets were overexpressed or knockdown of β FAAR for 48 h (12 islets well⁻¹). Thereafter the islets were washed and preincubated for 30 min at 37 °C in Krebs Ringer bicarbonate buffer (KRB), pH 7.4, supplemented with HEPES (10 mM), 0.1% bovine serum albumin, and 1 mmol/l glucose. After preincubation, the buffer was changed to a medium containing either 1 mmol/l or 16.7 mmol/l glucose. The islets were then incubated for 1 h at 37 °C. Immediately after incubation an aliquot of the medium was removed for analysis of insulin, and the islets were incubated in acid-ethanol for insulin content determination by human insulin ELISA kit (ExCell Bio, Shanghai, China), according to the manufacturer's instructions.

1. Wang, G. *et al.* Opposing effects of IL-1 β /COX-2/PGE2 pathway loop on islets in type 2 diabetes mellitus. *Endocrine journal* **66**, 691-699, (2019).

REVIEWERS' COMMENTS

Reviewer #2 (Remarks to the Author):

The authors have resolved my concerns by providing convincing new data. I have no further comments.